# Distinct insulin granule subpopulations implicated in the secretory pathology of diabetes types 1 and 2

Alex J B Kreutzberger[1,2†‡], Volker Kiessling[1,2†], Catherine A Doyle[3], Noah Schenk[4], Clint M Upchurch[3], Margaret Elmer-Dixon[1,2], Amanda E Ward[1,2], Julia Preobraschenski[5,6], Syed S Hussein[7], Weronika Tomaka[1,2], Patrick Seelheim[1,2], Iman Kattan[5], Megan Harris[8], Binyong Liang[1,2], Anne K Kenworthy[1,2], Bimal N Desai[1,3], Norbert Leitinger[3], Arun Anantharam[4], J David Castle[1,8]*, Lukas K Tamm[1,2]*

[1]Center for Membrane and Cell Physiology, University of Virginia, Charlottesville, United States; [2]Department for Molecular Physiology and Biological Physics, University of Virginia, Charlottesville, United States; [3]Department of Pharmacology, University of Virginia, Charlottesville, United States; [4]Department of Pharmacology, University of Michigan, Ann Arbor, United States; [5]Department of Neurobiology, Max Planck Institute for Biophysical Chemistry, Göttingen, Germany; [6]Cluster of Excellence in Multiscale Bioimaging: from Molecular Machines to Networks of Excitable Cells and Institute for Auditory Neuroscience, University of Göttingen, Göttingen, Germany; [7]Department of Microbiology, University of Virginia, Charlottesville, United States; [8]Department of Cell Biology, University of Virginia, Charlottesville, United States

*For correspondence:
jdc4r@virginia.edu (JDC);
lkt2e@virginia.edu (LKT)

[†]These authors contributed equally to this work

Present address: [‡]Department of Cell Biology Harvard Medical School and Program in Cellular and Molecular Medicine, Boston Children's Hospital, Boston, United States

Competing interests: The authors declare that no competing interests exist.

**Abstract** Insulin secretion from β-cells is reduced at the onset of type-1 and during type-2 diabetes. Although inflammation and metabolic dysfunction of β-cells elicit secretory defects associated with type-1 or type-2 diabetes, accompanying changes to insulin granules have not been established. To address this, we performed detailed functional analyses of insulin granules purified from cells subjected to model treatments that mimic type-1 and type-2 diabetic conditions and discovered striking shifts in calcium affinities and fusion characteristics. We show that this behavior is correlated with two subpopulations of insulin granules whose relative abundance is differentially shifted depending on diabetic model condition. The two types of granules have different release characteristics, distinct lipid and protein compositions, and package different secretory contents alongside insulin. This complexity of β-cell secretory physiology establishes a direct link between granule subpopulation and type of diabetes and leads to a revised model of secretory changes in the diabetogenic process.

## Introduction

Blood glucose levels are maintained in a narrow range by secretion of the hormone insulin from pancreatic β-cells. Insufficient insulin secretion leads to elevated levels of blood glucose resulting in diabetes (*Rorsman and Ashcroft, 2018*). Type 1 diabetes (T1D) results primarily from immune-mediated killing of pancreatic β-cells. In contrast, type 2 diabetes (T2D) arises when peripheral resistance to insulin signaling causes persistent demand for insulin secretion and eventual β-cell exhaustion (*Guthrie and Guthrie, 2004*; *Rorsman and Ashcroft, 2018*). Understanding molecular deficiencies of the secretory pathway in diabetes is essential for identifying the underlying causes of

**eLife digest** Diabetes is a disease that occurs when sugar levels in the blood can no longer be controlled by a hormone called insulin. People with type 1 diabetes lose the ability to produce insulin after their immune system attacks the β-cells in their pancreas that make this hormone. People with type 2 diabetes develop the disease when β-cells become exhausted from increased insulin demand and stop producing insulin.

β-cells store insulin in small compartments called granules. When blood sugar levels rise, these granules fuse with the cell membrane allowing β-cells to release large quantities of insulin at once. This fusion is disrupted early in type 1 diabetes, but later in type 2: the underlying causes of these disruptions are unclear.

In the laboratory, signals that trigger inflammation and molecules called fatty acids can mimic type 1 or type 2 diabetes respectively when applied to insulin-producing cells. Kreutzberger, Kiessling et al. wanted to know whether pro-inflammatory molecules and fatty acids affect insulin granules differently at the molecular level. To do this, insulin-producing cells were grown in the lab and treated with either fatty acids or pro-inflammatory molecules. The insulin granules of these cells were then isolated. Next, the composition of the granules and how they fused to lab-made membranes that mimic the cell membrane was examined.

The experiments revealed that healthy β-cells have two types of granules, each with a different version of a protein called synaptotagmin. Cells treated with molecules mimicking type 1 diabetes lost granules with synaptotagmin-7, while granules with synaptotagmin-9 were lost in cells treated with fatty acids to imitate type 2 diabetes. Each type of granule responded differently to calcium levels in the cell and secreted different molecules, indicating that each elicits a different diabetic response in the body.

These findings suggest that understanding how insulin granules are formed and regulated may help find treatments for type 1 and 2 diabetes, possibly leading to therapies that reverse the loss of different types of granules. Additionally, the molecules of these granules may also be used as markers to determine the stage of diabetes. More broadly, these results show how understanding how molecule release changes with disease in different cell types may help diagnose or stage a disease.

the disease. Secretory defects have been observed in diabetic human islet cells. Treatments that mimic these defects have been identified and used to model diabetes in healthy human cells, rodent islets, and immortalized cell lines (*Aslamy et al., 2018a*; *Aslamy et al., 2018b*; *Gandasi and Barg, 2014*; *Hoppa et al., 2009*; *Olofsson et al., 2007*). Onset of T1D is modeled by treatment of insulin-secreting cells with proinflammatory cytokines TNF-α, INF-γ, and IL-1β, which leads to decreased insulin release and a loss of Doc2B, an established biomarker for the disease (*Aslamy et al., 2018a*; *Aslamy et al., 2018b*). T2D is associated with extended exposure to elevated free fatty acids (FFA) caused by high fat diets (*Grill and Qvigstad, 2000*). The lipotoxicity component of T2D is frequently modeled by sustained treatment of insulin-secreting cells with palmitate, resulting in reduced glucose-stimulated insulin secretion (*Hoppa et al., 2009*; *Sako and Grill, 1990*).

Stimulated secretion of insulin normally occurs in two sequential phases, an acute first phase and a sustained second phase, which are differently affected in the T1D and T2D models. Modeling the inflammation of T1D by cytokine treatment causes a defect in insulin secretion attributed to the loss of Doc2B (*Aslamy et al., 2018a*; *Aslamy et al., 2018b*), which primarily suppresses the second phase of secretion (*Ramalingam et al., 2012*). Conversely, modeling T2D by palmitate treatment strongly decreases the first phase (*Rorsman and Ashcroft, 2018*). These observations show distinct insulin secretion profiles in the two diseases, and understanding the molecular and cellular mechanisms of how these different outcomes arise would provide a fundamentally new molecular view on the pathology of diabetes.

Although the secretory defects leading to T1D and T2D are well established, the cell and molecular biology that underlies the phenotypes of T1D and T2D model treatments is not known. To address this fundamental gap in knowledge, we performed large-scale purification of secretory granules on either untreated, palmitate-treated, or cytokine-treated insulin-secreting cells. These

granules were characterized comparatively for SNARE-mediated fusion properties in single particle fusion assays with SNARE-containing reconstituted target membranes. This novel cellular dissection and reconstitution approach, which was corroborated in intact insulin-secreting cells undergoing the same treatments, revealed a previously unknown heterogeneity among insulin granules with different subpopulations of secretory vesicles being lost following one or the other of the two diabetes-mimicking treatments. Further study showed remarkable differences in size, composition, and fusion characteristics of these subpopulations. As a result, we were able to identify a strong correlation between the distinct contents of the two subpopulations and known changes in β-cell output of signaling molecules by pancreatic islets during the onset of each type of diabetes. Taken together, these studies lead to a revised model about how the organization and regulation of two insulin secretory pathways potentially impact physiological intra-islet communication and pathological changes that accompany the diabetogenic process.

## Results

### Diabetic model treatments shift the calcium dependence of granule fusion

While secretion of insulin is known to be compromised in diabetes, we have been interested in exploring whether the diabetogenic process might entail detectable mechanistic changes in how insulin granules fuse during exocytosis. To address this question, insulin granules were purified by iso-osmotic density centrifugation (*Kreutzberger et al., 2019*) from INS 1 cells that stably express human proinsulin with GFP inserted into the C-peptide domain and that are referred to as GRINCH cells (*Haataja et al., 2013*) and used for biochemical characterization and reconstituted fusion assays. Binding and fusion of purified granules with planar supported membranes containing the SNARE proteins syntaxin-1a and SNAP-25 and lipids that mimic the plasma membrane in the presence of Munc18 and complexin were recorded by total internal reflection fluorescence (TIRF) microscopy in a reconstituted single particle fusion assay by monitoring the release of C-peptide-GFP (*Figure 1A*, *Figure 1—figure supplement 1A and D*). In this assay (*Kreutzberger et al., 2017a*), full-length syntaxin 1a and SNAP-25 are reconstituted into planar supported membranes and preincubated with Munc18 and complexin. Purified insulin granules are docked in the absence of calcium in a SNARE-specific manner (*Figure 1—figure supplement 1B*). In the absence of calcium, fusion is largely suppressed in the presence of Munc18 and complexin (*Figure 1—figure supplement 1C*). However, when calcium is injected, fusion occurs as monitored by a spike and decay in C-peptide-GFP TIRF fluorescence (*Figure 1—figure supplement 1D*). When triggered with 100 µM calcium, approximately 60% of bound granules fuse after a short time delay (*Figure 1—figure supplement 1D and E*) and the response is sensitive to the calcium concentration (*Figure 1—figure supplement 1F*).

Quantification of fusion as a function of calcium concentration revealed a biphasic response with granules purified from untreated cells (*Figure 1B*, green, control). Granules purified from cells following a 72 hr treatment with palmitate (delivered as a BSA complex and used to model elevated free fatty acid [FFA] observed during T2D *Rorsman and Ashcroft, 2018*) exhibited a monophasic high-affinity calcium response ($K_{1/2}$ = 12 ± 2 µM, *Figure 1B*, blue). In contrast, granules purified from cells following a 20 hr treatment with cytokines TNF-α, INF-γ, and IL-1β that model the inflammation associated with the onset of T1D (*Aslamy et al., 2018a*; *Aslamy et al., 2018b*) showed a shift to a low-affinity monophasic response ($K_{1/2}$ = 41 ± 4 µM, *Figure 1B*, red). Overexpression of Doc2B prior to cytokine treatment, which had previously been shown to reverse cytokine effects on β-cells (*Aslamy et al., 2018b*), prevented the shift of the calcium dose-response (*Figure 1C*, purple).

Fusion events of granules labeled with a soluble fluorescent content marker have a distinctive shape (*Figure 1D*, *Figure 1—figure supplement 1D*, *Figure 1—figure supplement 2* and see *Kreutzberger et al., 2017a*; *Kreutzberger et al., 2017b*; *Kreutzberger et al., 2019*). This is characterized by a decrease in fluorescence at the onset of fusion, a rise in fluorescence as the marker is pulled forward into the TIRF field as the granule membrane collapses into the supported membrane, and a continued decrease in fluorescence as the content probe diffuses away from the site of fusion. A two-step mathematical model can be used to describe the events in which the intensity change ($\Delta I_C$) indicates the amount of content released when the vesicle collapses into the supported

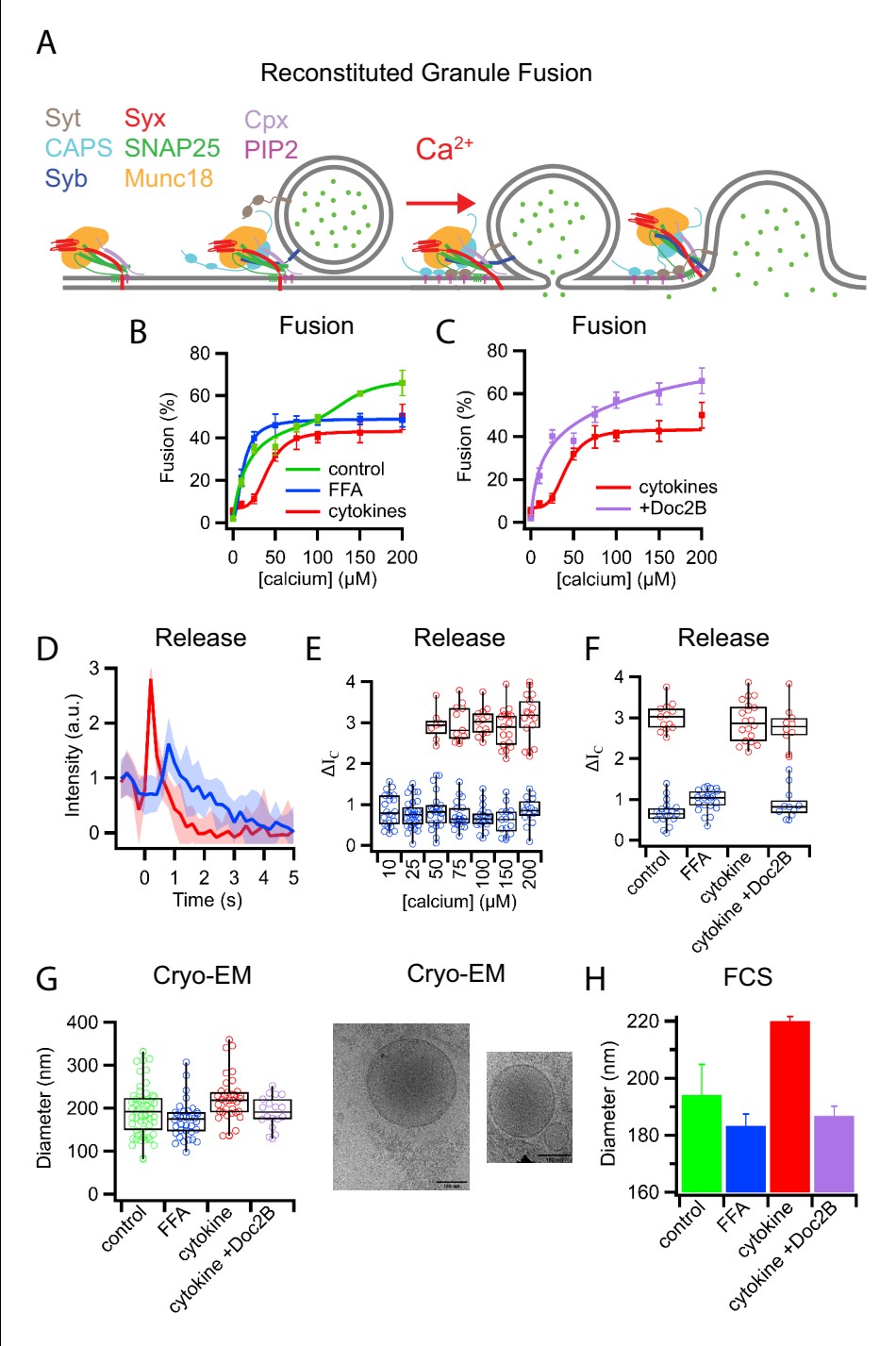

**Figure 1.** Palmitate (FFA) and cytokine treatments change the properties of insulin granules. (**A**) Schematic of components necessary for calcium-dependent reconstitution of insulin granule fusion with planar supported membranes (**Kreutzberger et al., 2019**). (**B**) Fusion of insulin granules purified from GRINCH cells as a function of calcium concentration for granules purified from untreated (green), palmitate-treated (blue), or cytokine-treated (red) GRINCH cells measured by TIRF microscopy. (**C**) Shift in fusion profile caused by cytokines (red) is prevented by the overexpression of Doc2B in GRINCH cells prior to cytokine treatment (purple). (**D**) Granule-to-planar supported membrane fusion events have slow (blue) or fast (red) release modes. Intensity traces are averages of 20 single events with the shaded region showing the standard deviation. (**E**) Quantification of intensity change $\Delta I_C$ (**Figure 1—figure supplement 2**) during fusion, for granules from untreated cells as a function of calcium showing the slow (blue) and fast (red) release modes. (**F**) The distribution of $\Delta I_C$ for granules purified from cells either

*Figure 1 continued on next page*

*Figure 1 continued*

untreated, palmitate-treated (FFA), cytokine-treated, or overexpressing Doc2B and cytokine-treated at 100 µM calcium. Numerical data of single granule fusion experiments are presented in Supplemental Data Tables. (**G and H**) Size distributions of purified insulin granules as determined by cryo-EM and FCS. Between 50 and 100 particles were measured in cryoEM micrographs from two different sample preparations per each condition in (**G**). The hydrodynamic radius was deduced from the FCS measurements (**H**) using the Stokes-Einstein relation (*Equation 4*) with correlation curves shown in *Figure 1—figure supplement 3*.

The online version of this article includes the following figure supplement(s) for figure 1:

**Figure supplement 1.** Characterization of insulin granule fusion with reconstituted planar supported membranes containing relevant SNARE proteins by TIRF microscopy.

**Figure supplement 2.** Granules release their content by distinct modes during fusion with planar supported membranes.

**Figure supplement 3.** Averaged FCS correlation curves along with curve fits.

---

membrane (*Figure 1—figure supplement 2*) and (*Kreutzberger et al., 2017a*; *Kreutzberger et al., 2017b*; *Kreutzberger et al., 2019*).

Averaging the fluorescent signals from multiple fusion events revealed that insulin granules had distinct slow and fast release characteristics for C-peptide-GFP release (*Figure 1D*). Quantifying individual granule fusion events from untreated cells showed slow release modes (smaller $\Delta I_C$) at all calcium concentrations (blue), while fast release modes appeared at higher calcium concentrations (*Figure 1E*). Granules purified following diabetic model treatments exhibited only slow release modes from palmitate-treated cells and only fast release modes from cytokine-treated cells when fusion was assayed at 100 µM calcium (*Figure 1F*). Overexpression of Doc2B restored the slow release mode of granules purified from cytokine-treated cells (*Figure 1F*).

Corresponding fast and slow release events were observed when analyzing secretion of C-peptide-GFP in intact GRINCH cells (*Figure 2* and *Figure 2—figure supplement 1*). Opening of voltage-dependent calcium channels increases intracellular calcium levels, which trigger granule exocytosis (*Rorsman and Ashcroft, 2018*). Increased depolarization, which increased calcium influx (*Figure 2—figure supplement 1A*), decreased the duration of C-peptide release events from intact cells (*Figure 2B and C* and *Figure 2—figure supplement 1B and C*) corresponding to fast and slow fusion events observed in the reconstitution assay (*Figure 1D*). At both high and low stimulation strengths, palmitate treatment increased release duration whereas cytokine treatment decreased release duration (*Figure 2C* and *Figure 2—figure supplement 1C*) corresponding to the shifts in release modes observed in the reconstitution assay (*Figure 1F*).

Purified insulin granules were further examined for treatment-induced morphology changes using cryo-EM (*Figure 1G*). Granules had a diameter of 195 ± 61 nm when purified from untreated cells, 173 ± 43 nm when purified from palmitate-treated cells, and 222 ± 53 nm when purified from cytokine-treated cells. Overexpression of Doc2B shifted the size distribution of purified granules back to 191 ± 37 nm. The shifts in granule size were confirmed using fluorescence correlation spectroscopy (FCS) to measure the hydrodynamic radii of the C-peptide-GFP containing granules with diameters of 194 ± 11 nm, 183 ± 4 nm, 220 ± 2 nm, and 187 ± 3 nm for granules from untreated, palmitate-treated, cytokine-treated, and Doc2B-protected cytokine-treated cells, respectively (*Figure 1H* and *Figure 1—figure supplement 3*). In combination, the data on calcium affinity, release modes, and granule size are consistent with the presence of two distinct subpopulations of insulin granules: larger fast releasing granules that are susceptible to palmitate T2D-mimicking treatment and smaller slow releasing granules that are susceptible to cytokine T1D-mimicking treatment.

## Synaptotagmin isoforms define the calcium affinity of distinct insulin granule subpopulations

The calcium sensor for exocytosis is synaptotagmin (syt), a granule membrane-associated protein with two C2 domains (*Brose et al., 1992*) that promotes assembly of the SNARE complex and catalyzes fusion between the granule and plasma membranes (*Jahn and Fasshauer, 2012*), leading to the release of insulin from the cell. Syt7 and syt9 are the two predominant isoforms relevant for insulin secretion (*Gustavsson et al., 2008*; *Iezzi et al., 2005*), with syt7 and syt9 known to have a high and low affinity for calcium, respectively (*Zhang et al., 2011*). The levels of both of these isoforms

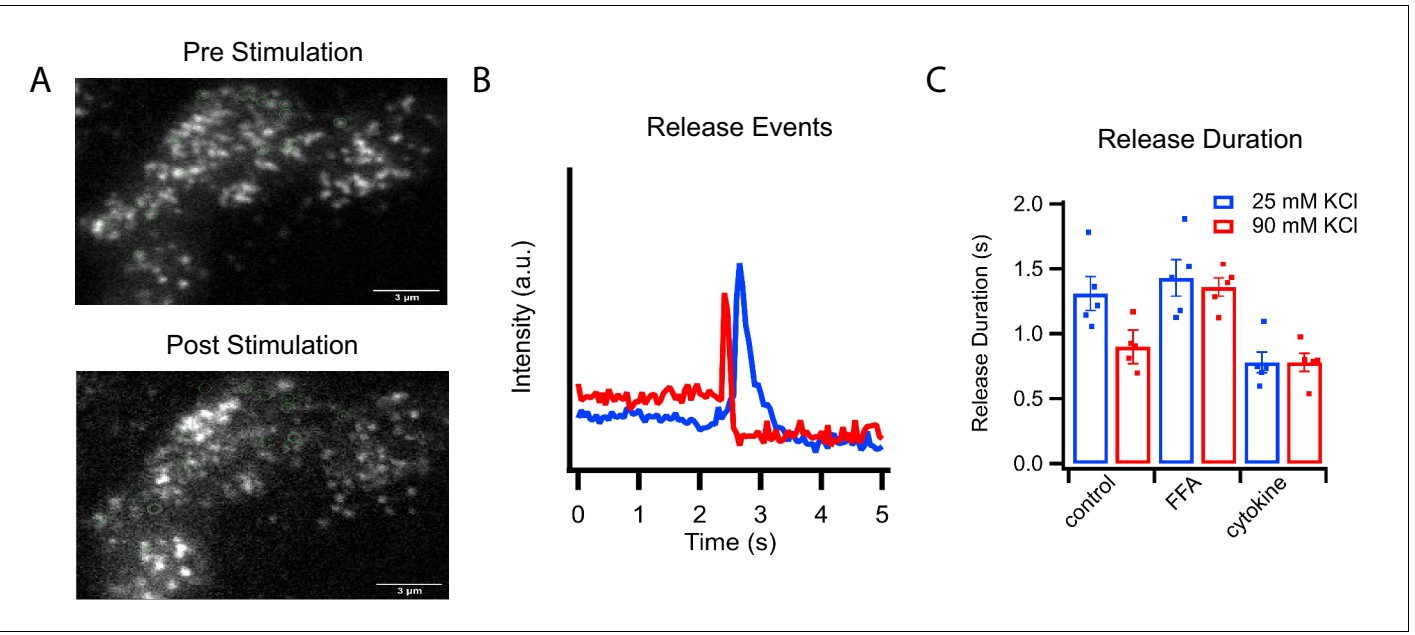

**Figure 2.** C-peptide-GFP release from GRINCH cells after KCl stimulation. (**A**) Representative images of a C-peptide-GFP labeled GRINCH cell before (top) and after stimulation (bottom); scale bar 3 μm. (**B**) Averaged intensity traces for single events of C-peptide-GFP exocytosis from GRINCH cells illustrating slower (blue) and faster (red) release durations. (**C**) Quantification of release duration for exocytotic events averaged over all events observed from each observed GRINCH cell following mild (25 mM KCl, blue) or strong (90 mM KCl, red) depolarization of control, palmitate- (FFA), or cytokine-treated cells. Distributions of individual exocytotic events are displayed in *Figure 2—figure supplement 1C*. Numerical data of cell secretion events are presented in *Figure 2—source data 1*. For statistics see *Figure 2—source datas 2* and *3*.

The online version of this article includes the following source data and figure supplement(s) for figure 2:

**Source data 1.** Numerical data of cell secretion events.
**Source data 2.** Significance values of release duration comparing stimulation strengths from *Figure 2C*.
**Source data 3.** Significance values of release durations comparing different cell treatments from *Figure 2C*.
**Figure supplement 1.** Calcium influx and single events of C-peptide-GFP release after KCl stimulation of GRINCH cells.

were unaffected in cell lysates after palmitate or cytokine treatment (*Figure 3A and B*). However, fractionation of cell lysates by density gradient centrifugation revealed that treatments of cells with T1D-mimicking cytokines or T2D-mimicking palmitate caused distinct and different shifts in syt isoform distribution (*Figure 3A and C*). Granules from palmitate-treated cells show a loss of the lower affinity syt9 while still containing the high-affinity calcium sensor syt7. Conversely, granules from cytokine-treated cells show a loss of syt7 while still containing syt9 (*Figure 3C*). Doc2B protected the redistribution of syt7 from fraction 9 of cytokine-treated cells (*Figure 3C*). Immunodepletion of intact granules from untreated cells using antibodies against either syt7 or syt9 revealed that these two isoforms reside on separate subpopulations of insulin granules (*Figure 3D*). The presence of distinct subpopulations is further supported by cryo-EM and FCS measurements on the immunodepleted supernatants. Residual syt7 granules (after syt9 depletion) had diameters of 168 ± 36 nm (cryo-EM) and 185 ± 4 nm (FCS), whereas residual syt9 granules (after syt7 immunodepletion) were 211 ± 46 nm (cryo-EM) and 197 ± 7 nm (FCS) (*Figure 3E* and *Figure 1—figure supplement 3*). We note that our iodixanol purified fraction 9 (*Figure 3C*) may contain a minor contamination from other organelles, but that this would not affect our results because we follow C-peptide-GFP, which is an authentic insulin granule marker.

Immunodepleted insulin granules were also examined by single particle fusion to reconstituted planar target membranes. The dose response to calcium was monophasic with a $K_{1/2} = 10 \pm 1$ μM for syt7 granules (depleted of syt9) and monophasic with a $K_{1/2} = 48 \pm 3$ μM for syt9 granules (depleted for syt7, *Figure 3F*). These results closely correlate with granules that were purified from PC12 cells engineered to express only the single respective syt isoform (*Figure 3—figure supplement 1*) and with the observation that syt isoforms give rise to different release rates in chromaffin

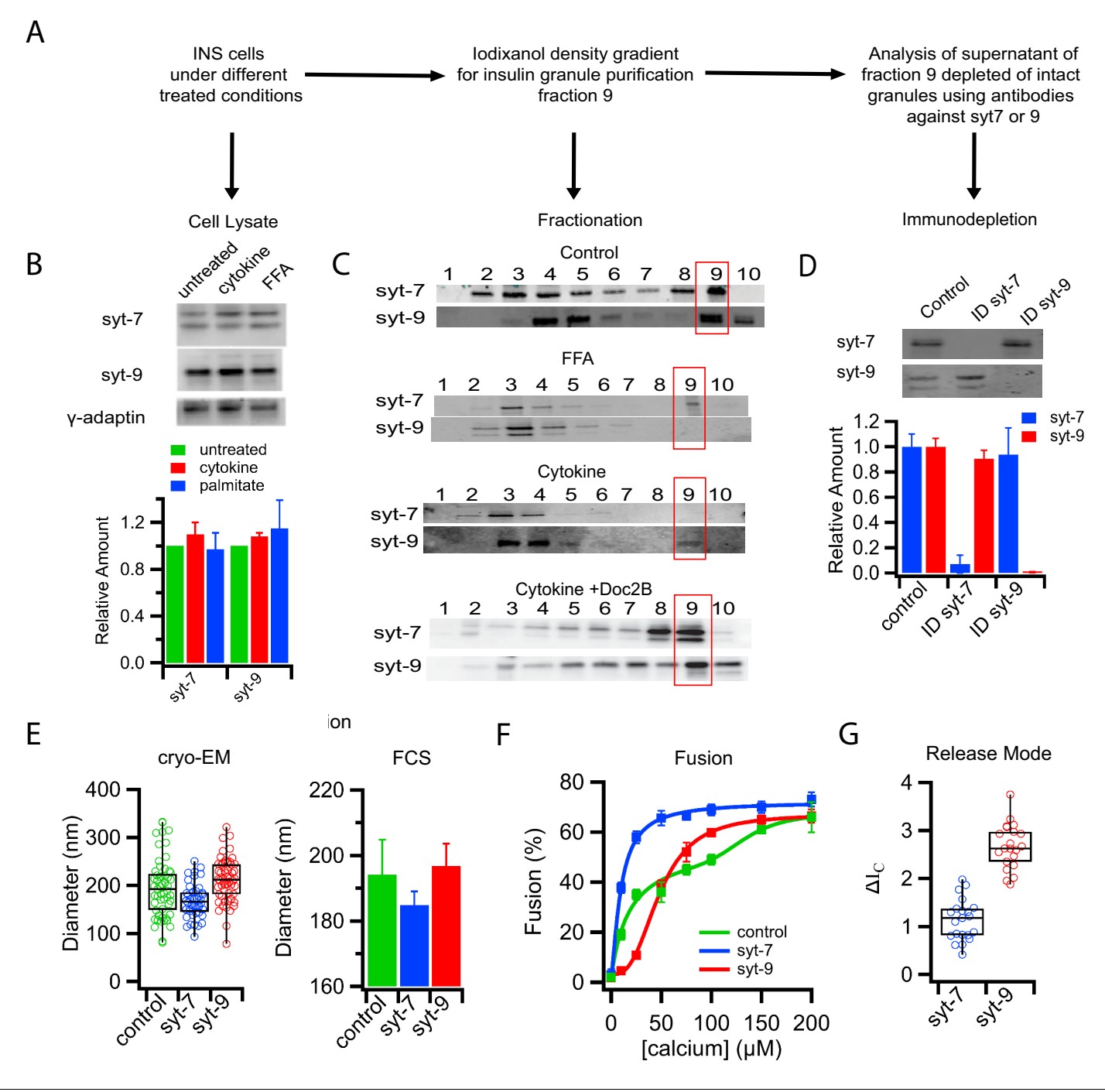

**Figure 3.** Subpopulations of insulin granules carry different synaptotagmin isoforms and have different physical and fusion properties that correlate with those observed after type-1 and type-2 diabetes model treatments. (A) Flow chart for purifying insulin granules from INS 1-derived (GRINCH or INS 832/13) cells over a density gradient, assessing their syt content, and further separation by syt isoform immunodepletion. (B) Western blots of cell lysates for either syt7 or syt9 from untreated cells or cells treated with cytokines and palmitate (FFA); accompanying quantification (from three independent experiments) shows that syt concentrations are unchanged. Gamma-adaptin was used for normalization as previously (*Hussain et al., 2018*). (C) Western blot of insulin granule fractionation profiles from cells following different treatments; fraction nine was collected as granules. (D) Western blot (top) and quantification (bottom) of insulin granule supernatants following immunodepletion with anti-syt7 or syt9 antibodies (n = 3). (E) Size distributions of total (control) and syt7 or syt9 granules harvested from supernatants following immunodepletion of the other granule type and measured by cryo-EM (left) or FCS (right). (F) Insulin granule fusion with planar supported membranes as a function of calcium concentration. Total (control) granules (green), syt7 granules (blue) and syt9 granules (red). (G) Content release mode (as defined in *Figure 1D* and *Figure 1—figure*

*Figure 3 continued on next page*

*Figure 3 continued*

supplement 2) for syt7 and syt9 insulin granules in the presence of 100 µM calcium. Numerical data of single granule fusion experiments are presented in *Figure 3—source data 1*.

The online version of this article includes the following source data and figure supplement(s) for figure 3:

**Source data 1.** Numerical data of single granule fusion experiments.
**Figure supplement 1.** Calcium dependence of granule fusion efficiency is determined by syt isoform.
**Figure supplement 2.** Release modes of PC12 dense core granules expressing only single synaptotagmin isoforms in a single vesicle/supported membrane fusion assay.

cells (*Rao et al., 2014*; *Rao et al., 2017*). Henceforth, insulin granule subpopulations will be referred to by the remaining syt isoform after depletion. Analysis of the modes of C-peptide-GFP release obtained from individual fusion events in the planar membrane fusion assay showed that the syt isoform also defines the release mode, with the syt7 granules releasing contents more slowly than the syt9 granules (*Figure 3G* and *Figure 3—figure supplement 2*). The similarity in shifts of calcium affinity, release mode, and size distribution lead us to conclude that T1D-mimicking cytokine treatment results in loss of syt7 granules, whereas T2D-mimicking palmitate treatment results in loss of syt9 granules.

## Granule subpopulations contain distinct lipid compositions

Formation of distinct insulin granule subpopulations strongly implicates a membrane sorting mechanism that begins in the trans-Golgi network (TGN) (*Simons and Ikonen, 1997*). Segregation potentially entails generation of granules with distinct lipid compositions analogous to those observed for other TGN-derived pathways (*Deng et al., 2016*). This was explored first by extracting lipids from all isolated granules (control), as well as from syt7 and syt9 subpopulations, and examining the relative amounts of cholesterol and sphingomyelin using colorimetric assays. Strikingly, syt7 granules were substantially enriched in both cholesterol and sphingomyelin compared to unseparated control granules, while syt9 granules contained much less of both lipid species (*Figure 4A* and *Figure 4—figure supplement 1*).

The marked difference in sphingomyelin partitioning between the two granule types was validated in both purified granule fractions and intact cells using a secretory pathway fluorescent biosensor, EQ-SM-Kate (*Deng et al., 2016*). EQ-SM-Kate consists of a mutated and nontoxic sphingomyelin-specific biosensor that is coupled to the fluorescent protein mKate and a signal sequence that directs entry into the secretory pathway. It was used in combination with a non-binding control EQ-sol-GFP encoding the same toxin further mutated so it does not bind membranes (*Deng et al., 2016*). mKate and GFP fluorescence of purified granules from INS 832/13 cells that had been co-transfected with both biosensors was measured before and after immunodepletion with antibodies against either syt7 or syt9 (*Figure 4B*). The GFP signal decreased in roughly equal amounts when immunodepleted of either syt isoform. In contrast, the SM-Kate signal only decreased when syt7 granules were depleted (*Figure 4B and C*). As further confirmation of differential labeling by EQ-SM-Kate, the distributions of EQ-SM-Kate and EQ-sol-GFP in transfected INS 832/13 cells were compared by confocal fluorescence microscopy. Punctate fluorescence signals were observed for both constructs (*Figure 4D*). The EQ-SM-Kate signal predominantly colocalized with the GFP signal while there were numerous GFP-positive puncta that lacked the EQ-SM-Kate signal (*Figure 4D and E*). This indicates that EQ-sol-GFP broadly labels vesicles in the secretory pathway while the SM-Kate sensor selectively labels organelles/granules enriched in sphingomyelin.

To further investigate other possible compositional differences of lipid species between granule populations, lipidomic analysis by mass spectroscopy of the phospholipids was performed. Purified granules divided into equal volumes were immunodepleted by syt isoform or left untouched (as a control). Then granule lipids were extracted and mass spectroscopy was performed. The relative abundance ratios of all detected lipid species in syt9 over syt7 granules were determined along with their significance values (*Figure 4—source data 2*) and plotted in a log-log volcano plot (*Figure 4F*). No detectable significant phospholipid species differences between the two granules were revealed with the exception of an enrichment of 42:1 sphingomyelin in syt7 granules with a significance value $p < 0.05$ (*Figure 4F*). When sphingomyelin 42:1, which likely contains the two predominant

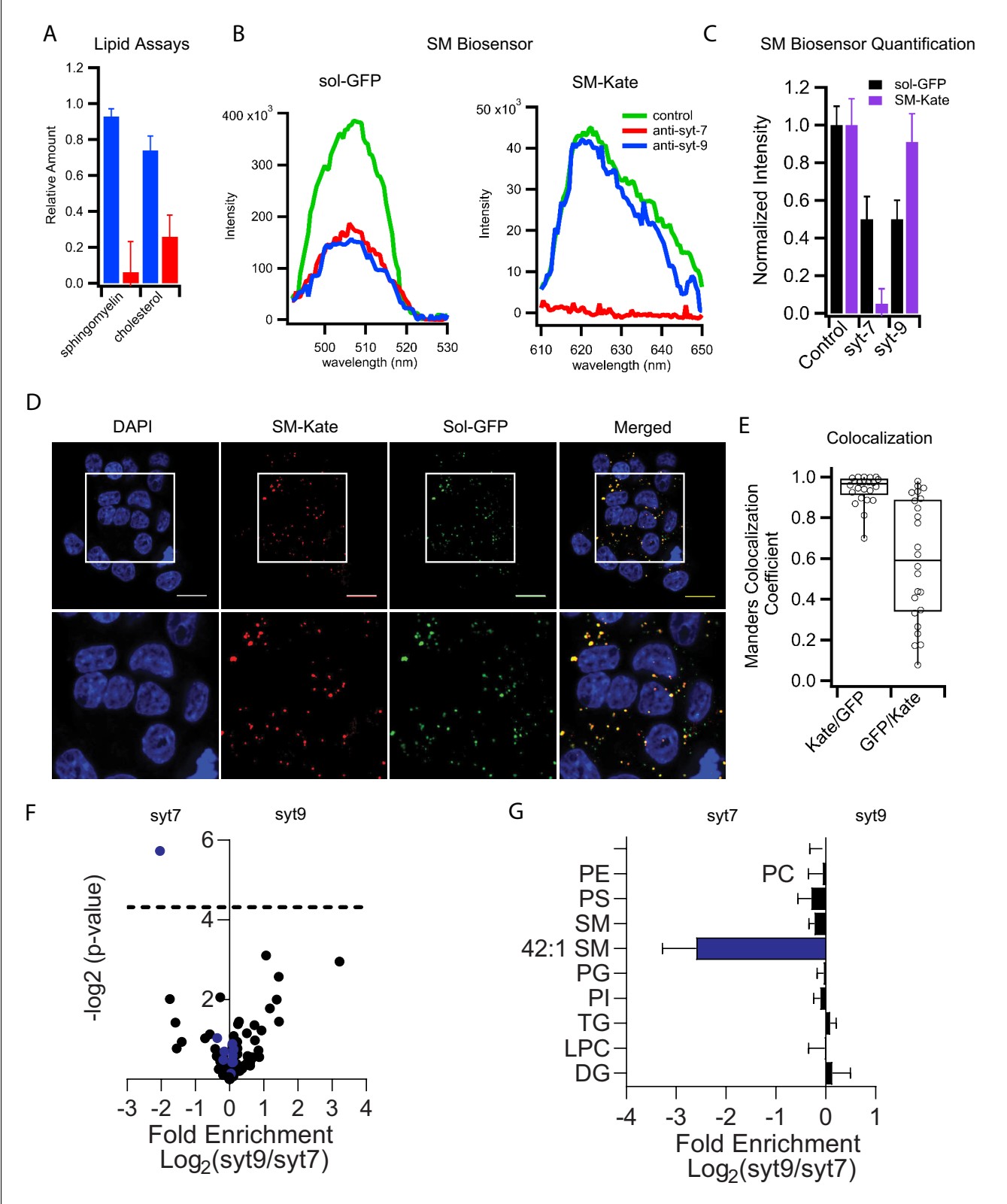

**Figure 4.** Syt7 and syt9 granule subpopulations have different lipid compositions. (**A**) Cholesterol and sphingomyelin contents of syt7 (blue) and syt9 (red) granules measured in lipid extracts using colorimetric assays (*Figure 4—figure supplement 1*, error bars are standard deviation from three independent samples). (**B**) Fluorescence spectra of GFP or mKate of intact granules in control and immunodepleted syt7 or syt9 granules purified from INS 832/13 cells. (**C**) Quantification of the fluorescence (sol-GFP in black and SM-Kate in purple) in control and immunodepleted syt7 or syt9 granules.
*Figure 4 continued on next page*

*Figure 4 continued*

Numerical data are presented in *Figure 4—source data 1*. (D) Fluorescence of INS 832/13 cells expressing plasmids that label all granules in the secretory pathway (sol-GFP) and those enriched in sphingomyelin (SM-Kate). Scale bars are 10 µm. (E) The Manders colocalization coefficient defining co-localization of SM-Kate to GFP (Red/Green) or GFP to SM-Kate (Green/Red). Each dot represents a single cell. (F) Volcano plot of mass spectrometry data comparing relative levels of specific lipid species from syt7 and syt9 granules. Data (from *Figure 4—source data 2*) are expressed as a ratio syt9/syt7 such that values to the right of 0 indicate enrichment in syt9 granules relative to syt7 granules and values to the left of 0 indicate the reverse. The significance of each measurement is plotted on the vertical axis. Blue dots are sphingomyelin species and black dots are all other lipid species. The only lipid showing a significantly different distribution (-log p-value>4.31 (dotted line), that is p<0.05) is SM 42:1, which is enriched in syt7 granules. (G) Bar graph derived from the data in panel F summarizing the compositional differences by lipid class where the fold enrichment in syt7 or syt9 granules is plotted, respectively, to the left and right of 0. At least three independent samples of syt7 and syt9 granules were analyzed by mass spectrometry (see *Figure 4—source data 3*).

The online version of this article includes the following source data and figure supplement(s) for figure 4:

**Source data 1.** Sphingomyelin biosensor data.
**Source data 2.** Table of lipid species, fold change, and significance of the change.
**Source data 3.** Lipidomics data.
**Figure supplement 1.** Lipid composition of INS 832/13 cell-derived insulin granule subpopulations.

---

sphingomyelin tails, 18:0 in the sphingosine and 24:1 in the acyl chain positions (*O'Brien and Rouser, 1964*), is visualized in a bar graph and compared to all other pooled lipid species in each headgroup class, it is apparent that sphingomyelin 42:1 is 3-fold enriched in syt7 compared to syt9 granules (*Figure 4G*). None of the other lipid classes and not even other sphingomyelin species show any significant enrichment in either type of granule.

## Palmitoylation of Syt7 correlates with selective association with sphingomyelin- and cholesterol-enriched granules

Previously, posttranslational palmitoylation of syt has been suggested to facilitate sorting to sphingomyelin- and cholesterol-enriched organelles. This has been shown for syt1 in neurons (*Kang et al., 2004*) and syt7 in immune cells (*Flannery et al., 2010*). To probe for this possibility in insulin secreting cells, palmitate containing a click-iT group was incubated with INS 832/13 cells to allow incorporation into endogenous proteins by posttranslational modification. Granules were purified from these cells, solubilized in detergent, and labeled with a corresponding click chemistry Alexa647 fluorescent dye. Subsequent immunoprecipitation with anti-syt antibodies revealed that syt7 but not syt9 was palmitoylated (*Figure 5A*). This likely explains the selective association of syt7 with sphingomyelin- and cholesterol-rich granules.

## Many proteins distribute selectively between granule subpopulations

β-Cells package proinsulin into nascent granules that bud from the TGN. Proinsulin is processed into insulin and C-peptide by granule-associated prohormone convertases PC1/3 and PC2. Insulin is then complexed with zinc and condensed into a crystalline core within mature granules. Stored granules are released by a signaling cascade, in which glucose uptake and subsequent metabolism increase the cellular ATP/ADP ratio, causing closure of plasma membrane ATP-dependent potassium channels and thereby depolarizing the cell.

The realization that there exist two subpopulations of insulin granules with distinct syt isoforms, fusion characteristics, and lipid compositions led us to examine other potential differences in content and membrane protein composition. Quantitative western blotting of immunodepleted granules was used to measure distributions between granule subpopulations of proteins that have functions in β-cell secretion or islet signaling and have established relationships to diabetes. The proteins considered include the membrane-anchored vesicular R-SNARE proteins, soluble SNARE-interacting proteins, non-insulin peptide hormones, their processing enzymes, and membrane transporters that concentrate classical transmitters within granules (*Figure 5B* and *Figure 5—figure supplement 1*).

The vesicular R-SNARE proteins, VAMP2, VAMP3, VAMP4, VAMP7, and VAMP8 are implicated in fusion of multiple post-Golgi trafficking pathways (*Dingjan et al., 2018*). VAMP2, VAMP3, VAMP4, and VAMP8 were found in roughly equal amounts in both syt7 and syt9 granules, whereas VAMP7 was significantly enriched in syt7 granules (*Figure 5B*). Of note, VAMP7 has been implicated in

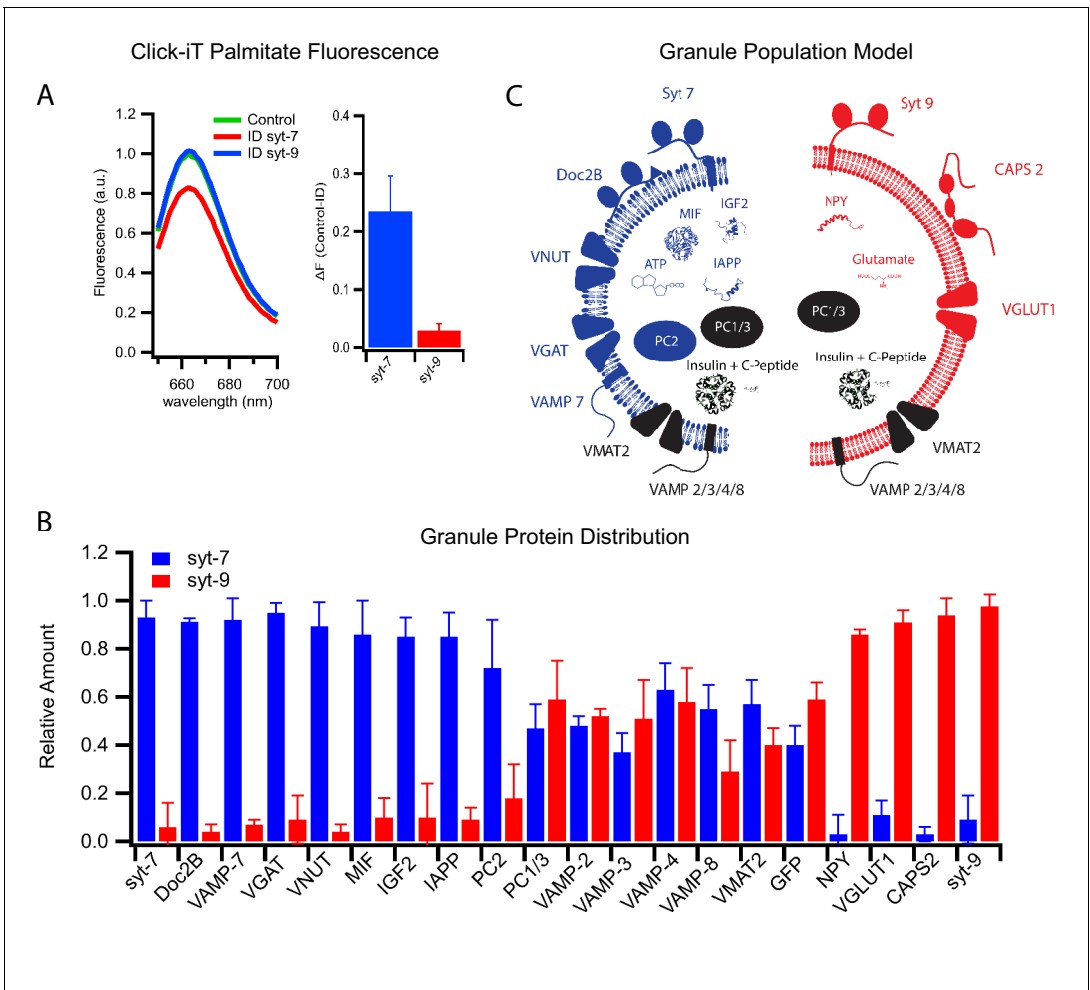

**Figure 5.** Differential protein distribution into syt7 and syt9 granule subpopulations. (**A**) Palmitic acid incorporates more efficiently into INS 832/13 cell-derived syt7 than into syt9 granules. Fluorescence spectra (left) of purified insulin granules with incorporated Click-iT palmitate solubilized in 1% Triton-X and then labeled with Alexa647 (by an Azide-Alkyne Click-iT reaction). The spectra are those of granules before (green) and after immunodepletion with syt antibodies (red, depletion using anti-syt7; blue, depletion using anti-syt9). Quantification of fluorescence change in control and syt7 or syt9 granules (right). The results are from three separate purifications of insulin granules. (**B**) Distribution of granule proteins between syt7 and syt9 granules that were derived from GRINCH cells. Data show the fraction of the total of each protein associated with syt7 (blue) vs syt9 (red) granules as determined from quantitative western blotting (*Figure 5—figure supplement 1*); error bars are standard deviation from three independent samples. For statistics see *Figure 5—source data 1*. (**C**) Model summarizing the compositions of syt7 (blue) and syt9 (red) insulin granules. The online version of this article includes the following source data and figure supplement(s) for figure 5:

**Source data 1.** Significance values of protein contents between different immunodepleted granules obtained from *Figure 5B* and *Figure 5—figure supplement 1*.

**Figure supplement 1.** Western blots for proteins associated with GRINCH cell-derived insulin granules before immunodepletion (left lanes) and after immundepletion of intact granules using antibodies against syt7 (middle lanes) or syt9 (right lanes).

crinophagy and macro-autophagy (*Csizmadia et al., 2018*), processes that are involved in insulin granule turnover (*Marsh et al., 2007*) and activated by the cytokine IL-1β (*Sandberg and Borg, 2006*).

Among upstream (of fusion) regulatory proteins calcium activated protein for secretion 2 (CAPS2) and Doc2B have been found to co-purify with the insulin granules (*Kreutzberger et al., 2019*). They each contain C2 calcium binding domains that interact with PI(4,5)P$_2$ and the SNARE fusion machinery (*Pinheiro et al., 2016*). CAPS2 functions in both granule maturation and priming for exocytosis (*Kreutzberger et al., 2017a*; *Speidel et al., 2008*), while Doc2B functions in priming likely through its recruitment of Munc13 (*Groffen et al., 2004*), a priming protein that does not co-purify with secretory granules (*Kreutzberger et al., 2019*). CAPS2 was found to be enriched on syt-9 granules,

while Doc2B was enriched on syt7 granules (*Figure 5B*), indicating that the fusion regulatory machinery differs in part between the granule subpopulations. Moreover, there is a striking correlation between the loss of syt7 granules and the loss of Doc2B with the cytokine-induced type 1 diabetic phenotype (*Figure 1B* and *Aslamy et al., 2018a*; *Aslamy et al., 2018b*). The preservation of the syt7 granules by overexpression of Doc2B suggests a role for Doc2B in granule formation or in protecting these granules during cytokine treatment.

Other β-cell secretory products implicated in the progression of diabetic phenotypes include neuropeptide Y (NPY), macrophage migration inhibitory factor (MIF), insulin-like growth factor 2 (IGF2), and islet amyloid polypeptide (IAPP). These secretory products enable local signaling in islets and distal regulation of other tissues. NPY promotes β-cell proliferation by inhibiting glucose-induced insulin secretion (*Rodnoi et al., 2017*). IGF2 promotes β-cell proliferation, and its autocrine action via the IGF1 receptor on β-cells (*Cornu et al., 2009*) has been shown to be elevated in T2D (*Casellas et al., 2015*). IAPP has been shown to inhibit insulin secretion and forms amyloid aggregates associated with T2D (*Westermark et al., 2011*). MIF increases β-cell insulin secretion via autocrine stimulation (*Waeber et al., 1997*) and suppresses islet resident macrophages (*Stojanovic et al., 2012*), an important signal for β-cell-to-immune system communication. Notably, IGF2, IAPP aggregates, and MIF have been shown to be elevated in patients with T2D, while MIF has been shown to be suppressed in T1D.

These secreted factors revealed a remarkable degree of segregation between the granule subpopulations. MIF, IGF2, and IAPP are detected in syt7 granules while NPY is predominantly detected in syt9 granules (*Figure 5B* and *Figure 5—figure supplement 1*). These findings are easily explained with the loss of syt7 granules in response to cytokines and their presence following extended palmitate treatment (*Figure 1B*). Evidently, our observed susceptibilities of distinct insulin granule subpopulations to the diabetes-mimicking treatments correlate well with previous clinical observations showing that the same respective granule specific markers are co-released with insulin in T1D and T2D patients (*Herder et al., 2006*). Overall, these results strongly support the notion that syt7 granules are lost selectively during early stages of T1D while syt9 granules are lost selectively in T2D.

The proprotein convertases PC1/3 and PC2 also exhibited distinct distributions. PC1/3 was equally distributed between the two granule subpopulations, while PC2 was markedly enriched in syt7 granules (*Figure 5B* and *Figure 5—figure supplement 1*). These differences suggest that processing specificity as well as processing kinetics may differ for the precursor polypeptides including insulin within the two granule types.

Secretion of small molecules including glutamate, GABA, ATP, and dopamine from β-cells plays a role in auto- and paracrine regulation of insulin secretion and in communication with pancreatic resident immune cells (*Bai et al., 2003*; *Gammelsaeter et al., 2004*; *Garcia Barrado et al., 2015*; *Geisler et al., 2013*; *Weitz et al., 2018*). These molecules are taken up and concentrated in insulin granules by specific transporters that use the electrochemical gradient generated by the VATPase. The vesicular transporters known to function in β-cells include VGLUT1 (glutamate uptake), VGAT (GABA uptake), VNUT (ATP uptake), and VMAT2 (dopamine uptake) (*Anne and Gasnier, 2014*). VMAT2 is equally distributed between the two subpopulations of granules. However, VGLUT1 is present only in syt9 granules, whereas VNUT and VGAT are present only in syt7 granules (*Figure 5B*). Thus, different neurotransmitters communicating with the immune system segregate into syt7 and syt9 granules. The distinct protein compositions of syt7 and syt9 granules are summarized in *Figure 5C*.

## Diabetic model treatments differentially affect secretion of products that partition between granule subpopulations

In order to verify the partitioning of different secretory products to granules with different syt isoforms and therefore different calcium affinities for granule exocytosis, secretion of three different products (C-peptide-GFP, used here as a surrogate for insulin and measured in a bulk secretion assay; ATP, measured by a luciferase assay of the cell media; and glutamate, measured in cell media by the iGluSnFr fluorescent biosensor) were examined using mild (25 mM KCl) and strong (90 mM KCl) depolarization eliciting weaker and stronger calcium responses, respectively (*Figure 2—figure supplement 1A*). Untreated GRINCH cells secreted C-peptide-GFP in increasing amounts as the depolarization strength was increased, consistent with its presence in both granule subpopulations (*Figure 6A*, top panel). ATP secretion measured with INS 832/13 cells was maximal at 25 mM KCl,

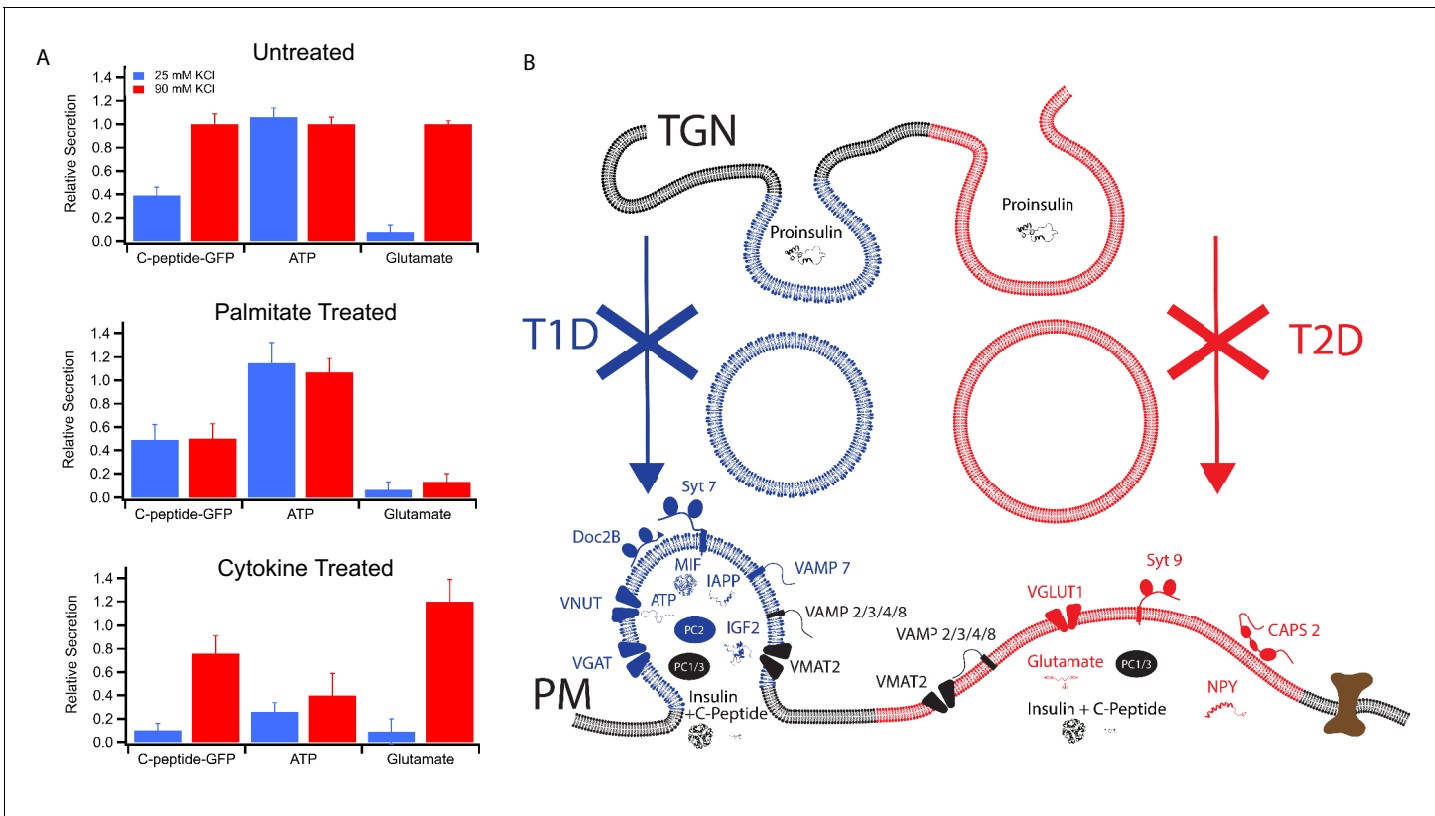

**Figure 6.** Secretion from GRINCH (C-peptide-GFP) or INS 832/13 (ATP and Glutamate) cells following treatments that model T1D and T2D phenotypes. (A) Response of secretory content from cells stimulated by 25 mM (blue) or 90 mM (red) KCl. C-peptide-GFP secretion was determined by a bulk secretion assay using GRINCH cells. ATP and glutamate were determined by a luciferase assay or the fluorescence of iGluSnFr glutamate biosensor, respectively, using INS 832/13 cells (error bars are standard deviations from three independent experiments). For statistics see *Figure 6—source data 1* and *2*. (B) Model of how different types of insulin granules are formed, released and affected in T1D and T2D. TGN, trans-Golgi network; PM, plasma membrane.

The online version of this article includes the following source data and figure supplement(s) for figure 6:

**Source data 1.** Significance values between different stimulation strengths for secretion experiments from *Figure 6A*.
**Source data 2.** Significance values between different treatments for secretion experiments from *Figure 6A*.
**Figure supplement 1.** Stimulation-dependent release of glutamate and ATP from INS 832/13 cells.

consistent with its selective concentration in syt7 granules. In contrast, glutamate secretion from INS 832/13 cells was only detected at 90 mM KCl, consistent with its selective concentration in syt9 granules (also see *Figure 6—figure supplement 1*, for calibrations and absolute amounts of secretion under different conditions). For palmitate-treated cells, C-peptide-GFP secretion was stimulated at 25 mM KCl and the response was not further increased at 90 mM KCl. Palmitate also had no observable effect on the secretion of ATP, but drastically reduced the secretion of glutamate (*Figure 6A*, middle panel). By contrast, the cytokine treatment reduced the secretion of C-peptide-GFP (especially upon mild stimulation) and significantly decreased the secretion of ATP, while secretion of glutamate was not much altered as compared to untreated cells (*Figure 6A*, bottom panel). Secretion of C-peptide, ATP, and glutamate from cells stimulated by strong (90 mM KCl) depolarization corroborates the calcium dose response for fusion of purified insulin granules (*Figure 1*) and the secretion by stimulated cells (*Figure 2*). The differential effects of T1D and T2D model treatments on C-peptide, ATP, and glutamate show a striking relationship between differential β-cell secretion and the pathology of diabetes.

The differing secretory effects of the diabetes-mimicking treatments potentially shed new insight regarding the onset and contrasting secretory changes for T1D and T2D. As summarized in *Figure 6B*, two subpopulations of granules containing either syt7 or syt9 are found in untreated cells. They have distinct affinities for calcium, exhibit distinct modes of content release, and they

have distinct lipid and protein compositions. Induction of a T1D model by treatment with cytokines results in loss of syt7 granules that normally secrete MIF, IGF2, IAPP, ATP, and GABA alongside insulin. On the other hand, induction of lipotoxicity in T2D by treatment with palmitate causes selective loss of syt9 granules, which normally release NPY and glutamate alongside insulin.

## Discussion

Our results reveal that insulin-secreting cells harbor distinct subpopulations of insulin granules that are selectively activated for secretion to release different secretory molecules. Therefore, insulin granules can no longer be considered generic storage units for releasing the full range of secretory molecules without regard to timing and signal strength. Regulated exocytosis provides a complex network of β-cell derived signals that serve as autocrine and paracrine governors of the secretion of insulin and other islet-derived hormones. The current work shows that these secretory regulators are parsed out selectively among the two insulin granule subpopulations that we have characterized in great molecular detail. The biphasic secretory pattern of insulin release in response to glucose has been previously described by its correlation to electrical activity in stimulated islet β-cells (*Rorsman and Ashcroft, 2018*). Because the granule subpopulations with secretory cargo compositions have different calcium affinities furnished by different synaptotagmin isoforms (*Figure 3*), we propose that these phases can be interpreted as a staged release of selected peptides and transmitters over time and levels of intracellular calcium. The detailed characteristics of the granule subpopulations and their correlations to T1D and T2D are summarized in *Table 1*.

In our new model linking the progression of T1D and T2D to different subpopulations of insulin secretory granules in the same cells, we posit that the syt9 granules release their content primarily in the first phase of secretion during the initial spike in intracellular calcium supported by their close proximity to calcium channels (*Hoppa et al., 2009*). Granules harboring the high-affinity calcium sensor syt7 would release their content during both phases. Indeed, both first- and second-phase secretion are decreased in human β-cells upon syt7 knockout (*Dolai et al., 2016*) and upon Doc2b knockout in mice (*Ramalingam et al., 2012*), while treatment of human β-cells with free fatty acids inhibits the peak during the initial release phase (*Hoppa et al., 2009*). It will be interesting to see if physiologic responses can be correlated to time-dependent release of selected cargoes that have local and systemic regulatory roles and if there is further heterogeneity within the discovered subpopulations.

Our studies further illustrate the very important contribution of granule membrane protein and lipid composition in supporting the complex signaling capability of β-cell exocytosis. The striking difference in lipid composition between the two granule subpopulations, especially with regard to sphingomyelin and cholesterol content (*Figure 4*), almost certainly contributes to the specific sorting of the fusion regulatory proteins and transporters (*Figure 5*). These sorting events enable granule subpopulations to be mobilized according to signaling strength and thereby export different regulatory signals during different phases of T1D and T2D development. Going forward, it will be interesting to define these sorting processes in more detail, determine how they interface with other post-Golgi pathways, and thereby contribute to the pathology of diabetes.

A central and striking new finding of the present study is the discovery that the two markedly different insulin granule subpopulations are each selectively sensitive to T1D and T2D model treatments. This brings new insight to the differing secretory pathologies that arise in β-cells during the two types of diabetes. Further, the differential loss of specific granule subpopulations can explain previously observed changes in insulin release characteristics such as more restricted fusion pores after palmitate treatment (*Hoppa et al., 2009*), which would be consistent with slower content release that we observed after this treatment. Because the levels of neither syt7 nor syt9 are decreased in lysates of cells following diabetic model treatments (*Figure 3B*), it appears that shifts in granule subpopulations induced by the treatments are not explainable solely by amplified autophagic degradation of a particular type of granule, although the process of granule turnover by crinophagy in β-cells (*Marsh et al., 2007*) may be membrane-conservative and thus not lead to synaptotagmin degradation. In the case of palmitate treatment, however, it will be necessary to evaluate this deduction in relation to the recently reported crinophagy of newly formed insulin granules under experimental conditions that differ from ours (*Pasquier et al., 2019*). Follow-up studies may also be needed to examine whether diabetogenic signaling is targeted to granule formation where a

**Table 1.** Characteristics of the granule subpopulations and their correlations to T1D and T2D.

| Characteristic | Syt7 subpopulation | Syt9 subpopulation | Action on or correlation to diabetes | Correlations in literature (references) |
|---|---|---|---|---|
| Calcium affinity ($K_{1/2}$) | $12 \pm 2$ μM | $41 \pm 4$ μM | | |
| Release rate | Slow | Fast | | |
| Size | ~180 nm | ~210 nm | | |
| Exocytosis regulatory | | | | |
| VAMP2 | + | + | | |
| VAMP3 | + | + | | |
| VAMP4 | + | + | | |
| VAMP7 | ++ | − | | |
| VAMP8 | + | + | | |
| CAPS2 | − | ++ | | |
| Doc2B | ++ | − | T1D (loss in human, rodent, culture models) | *Aslamy et al., 2018a*, *Aslamy et al., 2018b* |
| Prohormone processing | | | | |
| PC2 | ++ | − | | |
| PC1/3 | + | + | | |
| Molecular transporter | | | | |
| VGAT | ++ | − | GABA, immune communication | *Bai et al., 2003* |
| VNUT | ++ | − | GABA, immune communication | *Weitz et al., 2018* |
| VMAT2 | + | + | Dopamine, β-cell survival | *Garcia Barrado et al., 2015* |
| VGLUT1 | − | ++ | Glutamate, T1D onset | *Oresic et al., 2008* |
| Secretory/diabetes-related | | | | |
| MIF | ++ | − | T1D (decrease), T2D (increase)* | *Stojanovic et al., 2012* |
| IGF2 | ++ | − | T2D (increase)* | *Cornu et al., 2009*, *Casellas et al., 2015* |
| IAPP | ++ | − | T2D (amyloid aggregates increase)* | *Westermark et al., 2011* |
| NPY | − | ++ | T2D (progression) | *Rodnoi et al., 2017* |

*Note: for increases in T2D, the correlation is with being the only remaining granule subpopulation.

role for CAPS has already been defined (*Speidel et al., 2008*) or to enhanced stimulated secretion (*Olofsson et al., 2007*). As well, it will be interesting to learn whether perturbed clustering of L-type calcium channels by palmitate (*Gandasi and Barg, 2014*) and perturbed function of syntaxin-4 caused by proinflammatory cytokines (*Oh et al., 2012*; *Wiseman et al., 2011*) interface with the signaling that is targeted to the insulin granule subpopulations. Regardless, the loss of a whole granule subpopulation and with it the loss of a variety of signaling molecules during the onset of T1D or throughout T2D likely explains multiple previously unknown effects on the progression of the disease. For example, losing granules secreting ATP and MIF (*Figure 6A*) during the onset of T1D might compromise both local signaling and islet protection from immune cells (*Stojanovic et al., 2012*; *Weitz et al., 2018*).

Although there were prior hints that high and low sensitivity calcium pools co-exist in insulin secreting cells (*Wan et al., 2004*; *Yang and Gillis, 2004*), the identities of different granule subpopulations have not been previously described and mapping of the complex arrays of secretory products within these subpopulations has not been achieved. Also, the discovery that these subpopulations and their secretory products are linked to distinct diabetic states of insulin producing cells is novel and provides new mechanistic insight to established defects in diabetic (T1D and T2D) secretory signals. Our granule purification and functional reconstitution approach was ideally suited

to characterize heterogeneity of granules subpopulations in extensive detail, which has not been previously achieved using traditional cellular approaches.

A recent study demonstrates that the described granule heterogeneity likely extends to human and rodent pancreatic β-cells with differential modulation of calcium sensitivity of exocytosis conferred by synaptotagmin isoforms during pancreatic development and maturation (*Huang et al., 2018*). Therefore, our current work provides a road map for examining changes of secretory signals within diabetes in authentic islets. The new insights gained here will likely inform the design of future studies to explore granule heterogeneity in human and other mammalian islet secretion of specific products stimulated by appropriate signals under different diabetic states.

The presence of functionally and compositionally distinct secretory vesicles revealed here for insulin secreting cells (*Table 1*) is not unique to pancreatic β-cells and derivatives, but is likely a general property of a variety of regulated secretory cells. Secretory vesicle heterogeneity has been shown to have consequences on synaptic secretion and likely affects secretion from most endocrine cell types as well as from immune cells functioning in host defense. In neurons, the partitioning of VGAT and VGLUT between different vesicles has been observed (*Farsi et al., 2016*) but no mechanism of how a cell facilitates sorting of these transporters into distinct synaptic vesicles has been explored. The tools developed and presented in this work should open new avenues for future studies of the characteristics and mechanisms of sorting processes in these other cell types and thus help with a better molecular understanding of the diseases that are associated with them.

## Materials and methods

### Materials

The following materials were purchased and used without further purification: porcine brain L-α-phosphatidylcholine (bPC), porcine brain L-α-phosphatidylethanolamine (bPE), porcine brain L-α-phosphatidylserine (bPS), and L-α-phosphatidylinositol (liver, bovine) (PI), and porcine brain phosphatidylinositol 4,5-bisphosphate (bPIP$_2$) were from Avanti Polar Lipids; cholesterol, sodium cholate, EDTA, calcium, Opti-Prep Density Gradient Medium, sucrose, MOPS, glutamic acid potassium salt monohydrate, potassium acetate, adenosine 5'-triphosphate (ATP) magnesium (Mg$^{2+}$) salt and glycerol were from Sigma; CHAPS and DPC were from Anatrace; HEPES was from Research Products International; Click iT palmitic acid azide was from Molecular Probes; Sodium palmitate, Alexa Fluor 647 alkyne triethylammonium salt, chloroform, ethanol, Contrad detergent, all inorganic acids and bases, and hydrogen peroxide were from Fisher Scientific; fetal bovine serum (FBS) from Atlanta Biological. Water was purified first with deionizing and organic-free three filters (Virginia Water Systems) and then with a NANOpure system from Barnstead to achieve a resistivity of 18.2 MΩ/cm.

Antibodies for syt1 (mouse monoclonal cat. 105 011), syt7 (rabbit polyclonal cat. 105 173), syt9 (rabbit polyclonal cat. 105 053), Doc2B (rabbit polyclonal cat. 174 103), VGAT (mouse monoclonal cat. 131 011), VATPase (rabbit polyclonal cat. 109 002), VMAT2 (rabbit polyclonal cat. 138 302), VGLUT1 (mouse monoclonal cat. 135 011), CAPS2 (rabbit polyclonal cat. 262 103), synaptobrevin-2/VAMP2 (mouse monoclonal cat 104 211), and VAMP8 (rabbit polyclonal cat. 104 302) were from Synaptic Systems. The calnexin antibody (rabbit polyclonal cat. ADI-SPA-860) was from Enzo Life Sciences, the secretogranin II antibody (mouse monoclonal cat. LS-C335666) was from Biodesign International, the GFP antibody (mouse monoclonal cat. sc-9996) was from Santa Cruz, the succinate ubiquinone oxidoreductase antibody (mouse monoclonal cat. 459200) was from Molecular Probes, PC1/3 (rabbit polyclonal cat. Ab220363), PC2 (rabbit polyclonal cat. Ab3533), VAMP7 (rabbit polyclonal cat. Ab224535), NPY (rabbit polyclonal cat. Ab30914), IGF2 (rabbit polyclonal cat. Ab9574), and MIF (rabbit polyclonal cat. Ab7207) were from Abcam, IAPP (rabbit polyclonal cat. MBS8501010) was from MyBioSource, VNUT (rabbit polyclonal cat. Abn84) was from EMD Millipore, VAMP4 (rabbit polyclonal cat. PA1-768) was from Thermo Fisher, and VAMP3 (rabbit antiserum) was a gift from Pietro DeCamilli (Yale School of Medicine).

### INS 832/13 and GRINCH cell culture

INS 1 cell-derived 832/13 cells were obtained originally from Christopher Newgard, Duke University School of Medicine (*Hohmeier et al., 2000*). Cells were cultured on 10 cm plates in RPMI 1640 medium (Gibco), 10 mM HEPES, 1 mM sodium pyruvate, 50 μM β-mercaptoethanol, 1x pen/strep,

and 10% fetal bovine serum at 37°C and 5% $CO_2$ as described in *Hussain et al., 2018*. INS 832/13 cell-derived GRINCH cells stably expressing human proinsulin tagged with GFP in the C-peptide region, cultured with 20 µg/mL of G418 antibiotic, were originally described in *Haataja et al., 2013* and were a gift from Peter Arvan, University of Michigan Medical School. Cells were authenticated by immunofluorescent staining for endogenous insulin and the fact that a majority of our studies involved evaluating the insulin granule marker C-peptide-GFP. Potential contamination by myco-plasma was controlled by ciprofloxacin when needed. Free fatty acid (FFA) treatment with palmitate (sodium palmitate from Fisher, cat. P000725G) was performed as described by *Hoppa et al., 2009*. This was done by dissolving palmitate in 95% ethanol adding stoichiometric amounts of NaOH. The solution was dried with $N_2$ gas, water was added, and the sample was heated creating a hot soap. The solution was stirred and BSA was added to a final concentration of 10% w/v, creating a 10x stock solution. The pH was set to 7.4. Cells were cultured in media containing 0.5 mM palmitate in the presence of 1% BSA for 72 hr. This was estimated to give a FFA concentration of 26 nM (*Hoppa et al., 2009*). Cytokine treatments described by others (*Ahn et al., 2016*; *Aslamy et al., 2018b*) were performed by incubating 10 ng/mL TNF-α, 100 ng/mL of IFN-γ, and 5 ng/mL of IL-1β in cell culture media for 24 hr prior to experiments. Prior to cytokine treatment cells were trans-fected with (Myc-DDK-tagged)-human Doc2B (described in *Aslamy et al., 2018b* and purchased from OriGene, cat. RC218949). Transfection was performed by electroporation using a ECM 830 Electro Square Porator (BTX). After harvesting and sedimentation, cells were suspended in electro-poration buffer 120 mM KCl, 10 mM $KH_2PO_4$, 0.15 mM $CaCl_2$, 2 mM EGTA, 25 mM Hepes-KOH, 5 mM $MgCl_2$, 2 mM adenosine triphosphate, and 5 mM glutathione (pH 7.6) (*van den Hoff et al., 1992*) and then counted and diluted. Cell suspension of ~$10 \times 10^6$ cells in 700 µL and 30 µg of DNA were placed in an electroporator cuvette with a 4 mm gap, and two 255 V, 8 ms electroporation pulses were applied. Cells were transferred to a 10 cm cell culture dish with 10 mL of normal growth medium. Cytokine treatment was then applied 3 days after transfection.

## PC12 cell culture

Pheochromocytoma cells (PC12) were cultured on 10 cm plates in DMEM high-glucose media (Gibco) supplemented with 5% horse serum (CellGro), and 5% calf serum (HyClone), and 1% pen/strep. Stable knockdowns of endogenous synaptotagmins (isoforms 1 and 9) were generated (first described in *Kreutzberger et al., 2017a*) individual isoforms were over expressed with shRNA resis-tant plasmids for synaptotagmin-1,–7, or −9 (*Bendahmane et al., 2020*; *Kreutzberger et al., 2017a*; *Kreutzberger et al., 2019*).

## GRINCH and PC12 cell granule purification

Granules were purified using a previously described method (*Kreutzberger et al., 2017a*; *Kreutzberger et al., 2019*). Cells were scraped (~20–30 10 cm plates) into PBS. Cells were pelleted by centrifugation and then suspended and washed in homogenization medium (0.26 M sucrose, 5 mM MOPS, 0.2 mM EDTA) by pelleting and resuspending. Following resuspension in 3 mL of medium containing protease inhibitor (Roche Diagnostics), the cells were cracked open using a ball bearing homogenizer with a 0.2507 inch bore and 0.2496 diameter ball. The homogenate was spun at 1500 x g for 10 min at 4° C in fixed-angle microcentrifuge to pellet nuclei and larger debris. The post-nuclear supernatant was collected and spun at 5900 x g for 15 min at 4°C to pellet mitochon-dria. The post-mitochondrial supernatant was then collected, adjusted to 5 mM EDTA, and incu-bated 10 min on ice. A working solution of 50% Optiprep (iodixanol) (5 vol 60% Optiprep: 1 vol 0.26M sucrose, 30 mM MOPS, 1 mM EDTA) and homogenization medium was used to prepare solu-tions for discontinuous gradients in Beckman SW55 tubes: 0.5 mL of 30% iodixanol on the bottom and 3.8 mL of 14.5% iodixanol, above which 1.2 ml EDTA-adjusted supernatant was layered. Sam-ples were spun at 190,000 x g for 5 hr. A clear white band at the interface between the 30% iodixa-nol and the 14.5% iodixanol was collected as the dense core granule sample. Gradient fractions were prepared for western blotting by diluting individual fractions with PBS, followed by pelleting the membranes with a high-speed spin and resuspending the membranes in SDS loading buffer. The dense core granule sample used for functional experiments was extensively dialyzed in a cassette with 10,000 kD molecular weight cutoff (24–48 hr, 3 × 5L) into the fusion assay buffer (120 mM potassium glutamate, 20 mM potassium acetate, 20 mM Hepes, pH 7.4).

## Western blots

Samples were suspended in 1X SDS buffer. Samples were loaded and run on 4–20% TGX protein gels (Bio-Rad 4561094). Gels were then transferred onto a PVDF membrane with Bio-Rad Trans-Blot Turbo Transfer system (Bio-Rad 1704150); dried membranes were activated with methanol and washed in TBST (0.5% Tween-20 in Tris-buffered saline) before use. Membranes were blocked in 5% milk in TBST for 1 hr, washed, and incubated in primary antibody overnight. All primary and secondary antibodies for immunoblotting were diluted in Millipore signal boost immunoreaction enhancer solutions (Millipore 407207). The following day, membranes were washed and probed with HRP conjugated secondary antibody diluted 1:10,000 (Jackson Immunoresearch cat. rabbit: 211032171, cat. mouse: 115035174). Membranes were washed and incubated in Pico PLUS chemiluminescent substrate (Thermofisher 34580) before imaging on Fujifilm LAS-3000 luminescent analyzer.

## Immunodepletions of synaptotagmin isoforms

100 µL of Protein A magnetic SureBeads (Bio-Rad) were washed three times with PBS with 1% Tween, three times with PBS, and then washed five times with fusion buffer (120 mM potassium glutamate, 20 mM potassium acetate, 20 mM HEPES, pH 7.4). The beads were then incubated with 5 µg of the indicated synaptotagmin isoform antibody in 100 µL of fusion buffer for 20–30 min. Beads were then washed three times with fusion buffer. Antibody-bound beads were then incubated with 0.5–1 mL of purified insulin granules (sufficient to adsorb all granules containing an individual synaptotagmin isoform; *Figure 3D*). Supernatant was used for fusion experiments or was pelleted and resuspended in SDS loading buffer for western blots. Western blots of immuno-depleted granules (*Figure 5—figure supplement 1*) were quantified with the amount of protein in depleted samples being determined relative to the amount of an equal volume of granules that were not depleted (*Figure 5B* and *Figure 5—figure supplement 1*).

## TIRF-based single granule cell secretion assay

Imaging was performed on an Olympus cellTIRF-4Line microscope (Olympus, USA) equipped with a TIRF oil immersion objective (NA 1.49) and an additional 2x lens in the emission path between the microscope and the cooled electron-multiplying charge-coupled device Camera (iXon 897, Andor Technology). The final pixel size was 80 nm. Series of images were acquired at ~20 Hz using Cell-Sense software with an exposure time of 30 ms and an EM gain of 200. C-peptide-GFP was excited using a 488 nm laser. The cells were imaged in buffer (145 mM NaCl, 5.6 mM KCl, 2 mM CaCl$_2$, 0.5 MgCl$_2$, 3 mM glucose, and 15 mM HEPES, pH 7.4). Cells were individually stimulated using a needle (100 µm in diameter) connected to a perfusion system under positive pressure ALA-VM4 (ALA Scientific Instruments, Westerbury, NY). To trigger exocytosis, cells were first perfused with buffer for 5–10 s and then stimulated by increasing KCl to 25 mM or 90 mM with a subsequent decrease in NaCl by equal amounts. Images were analyzed using a homemade program written in LabVIEW (National Instruments). Stacks of images were filtered by a moving average filter. The maximum intensity for each pixel over the whole stack was projected onto a single image. Insulin granules were located in this image by a single particle detection algorithm (*Kiessling et al., 2006*). The central pixel of fluorescence intensities of a 5-pixel by 5-pixel area around each identified center of mass were plotted as a function of time for all particles in the image series. The exact time points of the onset of fusion until the content release completed of granules docked prior to stimulation or granules that came into the TIRF field after stimulation was determined. Some granules moved after docking and before fusion. These were not included in the analysis of fast vs. slow fusion granules.

## Calcium imaging

96-well glass bottom plates (Cellvis P96-0-N) were coated with 0.1% Poly-L-Lysine (Sigma P8920) and washed with sterile water. GRINCH cells were detached from cell culture plates with 0.25% Trypsin-EDTA (Gibco 25200–056), washed with culture medium, and pelleted at 500 g for 5 min. Cells were resuspended in Fluo-4 AM solution containing buffer (155 mM NaCl, 4.5 mM KCl, 1 mM MgCl$_2$, 2 mM CaCl$_2$, 5 mM HEPES, 10 mM Glucose, pH = 7.4), 5 uM Fluo-4 AM (ThermoFisher F14201), 0.02% Pluronic F-127 (ThermoFisher P3000MP), and 500 µM Probenecid. Cells were plated at a density of 200,000 cells/well, pelleted at 500 g for 2 min, and incubated at room temp for 30 min covered from light. Fluo-4AM solution was removed and replaced with 100 µL of buffer

containing 500 μM Probenecid and incubated for 5 min covered from light. Immediately before imaging each column, buffer with probenecid solution was removed and cells were washed with buffer without Probenecid. 80 μL of buffer was added and cells were placed in Flexstation three for recording. Each column of the 96-well plate was recorded separately. Calcium flux was measured on Flexstation three at excitation of 490 nm and emission of 520 nm every 2 s for 360 s at 25C and recorded on SoftMax Pro 7.03 software. At 31 s, 20 μL of buffer containing 4.5 mM, 125 mM, or 450 mM KCL was added to appropriate wells for final concentrations of 4.5 mM (baseline), 25 mM, or 90 mM KCl respectively. At 300 s 25 μL of 1% Triton (0.2% final concentration) was added to permeabilize cells and thus expose Fluo-4 AM to 2 mM $Ca^{2+}$ containing buffer. This was used as the positive control. Data are reported as F1-F0 normalized to positive control.

## Bulk C-peptide-GFP secretion assay

Secretion of C-peptide-GFP was performed as previously described (*Hussain et al., 2018*). GRINCH cells were seeded at the same density and cultured in Roswell Park Memorial Institute (RPMI) media with 10% FBS. The cells were washed into 1 mL of basal buffer (145 mM NaCl, 5.6 mM KCl, 2.2 mM $CaCl_2$, 0.5 mM $MgCl_2$, 3 mM glucose, 15 mM HEPES, pH 7.4) and incubated at 37°C for 30 min. Cells were then stimulated by increasing the KCl concentration to 25 or 90 mM (with a corresponding decrease in NaCl). The supernatant was collected and any cellular debris was pelleted. Cells were scraped, pelleted, and lysed and the supernatant was used to assay GFP fluorescence. Fluorescence at 510 nm (emission peak of GFP) recorded in a SpectraMax M5 plate reader (Molecular Devices) and was used to calculate fractional secretion (percent of total GFP fluorescence secreted under stimulated conditions relative to the total amount of GFP in cell lysates).

## Glutamate secretion assay

INS 832/13 cells were plated at equal density per plate and allowed to grow for 24 hr before treatments were performed. The cells were washed into 1 mL of basal buffer and incubated at 37°C for 5 min. The buffer was then collected and cells were incubated with 1 mL of 25 mM KCl or 90 mM KCl stimulation solution for 3 min 37°C. The collected stimulation media were centrifuged to pellet any residual cells. Cells were scraped, pelleted, and lysed for the protein concentration to be determined by a BCA assay (BCA Protein Assay Kit from Thermo Fisher Scientific cat. 23227). The amount of glutamate was then measured in 100 μL of the media containing the iGluSnFr.A184S fluorescence glutamate sensor variant (*Marvin et al., 2013*; *Marvin et al., 2018*) using a Flexstation three at excitation of 490 nm and emission of 520 nm at 25°C and recorded using SoftMax Pro 7.03 software. The fluorescence was normalized to the amount of protein in the cell lysates to account for any difference in cell density. The relative secretion (*Figure 6A*) was determined by normalizing each sample to the amount of secretion observed for untreated cells at 90 mM KCl.

## ATP secretion assay

INS 832/13 cells were plated at equal density per plate and allowed to grow for 24 hr before treatments were performed. The cells were washed into 1 mL of basal buffer and incubated at 37°C for 5 min. The buffer was then collected and cells were incubated with 1 mL of 25 mM KCl or 90 mM KCl stimulation solution for 3 min at 37°C. The stimulation media were centrifuged to pellet any cells. Cells were scraped, pelleted, and lysed for the protein concentration to be determined by a BCA assay (BCA Protein Assay Kit from Thermo Fisher Scientific cat. 23227). The amount of ATP was assayed using a luminescent ATP detection assay (Abcam) with bioluminescence being recorded on Flexstation 3 at 25°C and recorded using SoftMax Pro 7.03 software. The luminescence was normalized to the amount of protein in the cell lysates to account for any difference in cell density. The relative secretion (*Figure 6A*) was determined by normalizing each sample to the amount of secretion observed for untreated cells at 90 mM KCl.

## Click-iT palmitoylation assay

Cells were grown to ~70% confluency and then incubated with 50 μM Click-iT palmitic acid azide (Molecular Probes cat. C10265) for 6 hr (10 plates of cells were used). Cells were then scraped into PBS and insulin granules were purified. Granules were dialyzed for 36 hr in buffer (120 mM potassium glutamate, 20 mM potassium acetate, and 20 mM HEPES, pH 7.4) with two buffer changes.

The purified granules where solubilized using 1% Triton-X100 detergent and then the Click-iT azide group was allowed to react with an Alexa Fluor 647 Alkyne (Thermo Fisher Scientific cat. A10278) for 1 hr. After reaction, the granules were split into three samples. One sample was left untreated while the other two were immunodepleted using either anti-syt7 or syt9 antibodies. The fluorescence of the supernatants was then compared to the control for loss of Alexa647 fluorescence. This experiment was repeated with three independent granule purifications.

## Determination of cholesterol and sphingomyelin content

Samples of each biological replicate (three replicates in total) were split into three equal volumes. One sample was left untouched as a control, while the other two were immunodepleted for one or the other syt isoform. Following immunodepletion the amount of protein per sample was determined by a BCA assay (BCA Protein Assay Kit from Thermo Fisher Scientific cat. 23227), cholesterol was quantified using a colormetric assay (Amplex Red Cholesterol Assay from Thermo Fisher Scientific, cat. A12216), and sphingomyelin was quantified using an fluorometric assay (Sphingomyelin Assay Kit from Abcam, cat. ab138877). The amount of cholesterol or sphingomyelin per amount of protein in the sample was determined for control and immunodepleted samples (*Figure 4—figure supplement 1*). The relative amounts of cholesterol and sphingomyelin as compared to the control sample are shown (*Figure 4A*) by normalizing the immunodepleted samples to equal amounts of samples that were not immunodepleted.

## Sphingomyelin-biosensor assays

To purify granules labeled with EQ-sol-GFP and EQ-SM-Kate (gifts of Christopher Burd, Yale School of Medicine), cells were simultaneously transfected with both constructs using an ECM 830 Electro Square Porator (BTX, Hawthorne, NY). Accordingly, cells were harvested and suspended in 0.75 mL of a sterile cytomix electroporation buffer (120 mM KCl, 10 mM KH$_2$PO$_4$, 0.15 mM CaCl$_2$, 2 mM EGTA, 25 mM Hepes-KOH, 5 mM MgCl$_2$, 2 mM ATP, and 5 mM glutathione, pH 7.6) (*van den Hoff et al., 1992*). Cells were then diluted to $10 \times 10^6$ cells in 700 µL buffer with 30 µg of DNA and electroporated in a cuvette with a 4 mm gap and two 255 V 8 ms electroporation pulses were applied. Subsequently, cells were transferred to a 10-cm cell culture dish with 10 mL of normal growth medium and grown for 4 days before insulin granules were purified. Purified insulin granules were dialyzed for 36 hr (two buffer changes) with fusion buffer (120 mM potassium glutamate, 20 mM potassium acetate, and 20 mM Hepes, pH 7.4) to remove iodixanol. Dialyzed granules were divided into three samples of equal volume – one left untreated, one immunodepleted for synaptotagmin-7, and one immunodepleted for synaptotagmin-9. After immunodepletion the fluorescence spectra of the supernatants were recorded on a Fluorolog 3 (model FLS-21 from Horiba) and compared to the spectrum of the untreated sample.

For fluorescence microscopy, cells were plated at 80% confluency in a 6-well plate. Then 2.5 µg of EQ-sol-GFP and EQ-SM-Kate plasmids were transfected with lipofectamine 3000 (ThermoFisher Cat #L300015) into each well. Cells were incubated for 3 days at 37°C, plated onto cover slips, fixed in 3% formaldehyde for 10 min (Cell Signaling Technology #12606) and mounted onto slides with DAPI stain (Cell Signaling Technology #8961). Cells were then imaged on a Zeiss LSM-880 confocal microscope. Image analysis for colocalization of EQ-sol-GFP and EQ-SM-Kate was quantified using the coloc2 plugin in ImageJ. Each cell region of interest was manually defined and Manders Colocalization Coefficient was generated to quantify the degree of colocalization of EQ-sol-GFP and EQ-SM-Kate puncta. M1 Red/Green defines the extent of EQ-SM-Kate colocalizing with EQ-sol-GFP. M2 Green/Red represents the extent of sol-GFP colocalizing with SM-Kate.

## Mass spectrometry

Lipids were isolated from syt7 and syt9 granules purified from INS 832/13 cells using a modified Bligh-Dyer method. In brief, 400 µL of granules in PBS were added to 500 µL of CHCl$_3$ (EMD Cat. CX1050-1) and 1 mL methanol (Sigma-Aldrich Cat. 34860) containing Splash Lipidomix Mass Spec Standard (1 µL, Avanti Polar Lipids, Inc, Cat. 330709) and centrifuged for 5 min at 787 g (Eppendorf Centrifuge 5810 R). Samples were decanted and 600 µL of water (HPLC grade, Sigma-Aldrich Cat. 270733) and 1.5 mL of CHCl$_3$ were added to each sample. Samples were vortexed and centrifuged (5 min at 787 g). The organic layer was collected. The extraction was repeated for a total of three

extractions. The combined organic layers were dried under nitrogen gas and resuspended in 400 μL of methanol. Samples were analyzed using a ThermoFisher Q Exactive mass spectrometer coupled with a Vanquish Ultra-High Performance Liquid Chromatography system. Chromatographic separation was achieved using a 10–100% gradient of Solvent A over 24 min at a constant flow rate of 400 μL/min (Solvent A: 50% acetonitrile (Fisher Scientific Cat. A998-4) in water, 10 mM ammonium formate (Acros Organics Cat. 401152500) and 0.1% formic acid (Fluka Cat. 56302); Solvent B: 10% acetonitrile, 88% 2-propanol (Sigma-Aldrich Cat. 34863), 2% water, 2 mM ammonium formate, and 0.02% formic acid) on a ThermoFisher Acclaim 120 C18 column (5 μM, 120 Å, 4.6 × 100 mm). Lipid species masses were collected by ddMS$^2$ Top5 with an inclusion list for Splash Lipidomix Mass Spec Standard lipid species. Lipid identities were assigned by LipidSearch 4.1.16. The resulting species were normalized to d18:1-18:1(d9) sphingomyelin and protein concentration.

To identify lipid *species* that were significantly enriched in either syt7 (immunodepleted of syt9) or syt9 (immunodepleted of syt7) granules, sample replicates for each lipid species were averaged and the ratio (syt9/syt7) was calculated. The ratios for each lipid species were log$_2$ transformed. Significance for each lipid species was calculated by two-tailed Student's T-test with unequal variance. The resulting p-values were log$_2$ transformed and -log$_2$ significance vs. log$_2$(syt9/syt7), which represents fold enrichment in syt9 over syt7 granules, was plotted to generate a Volcano plot. A p-value of 0.05 or lower (corresponding to -log$_2$ > 4.31) is considered a significant change in lipid content between the two types of granules.

To identify lipid *classes* that were enriched in either syt7 or syt9 granules, the peak areas for each class were totaled on a per sample basis. The resulting lipid class totals for each sample were normalized to the average of syt7 granules for each lipid class. The normalized data was then log$_2$ transformed and the data represented in a bar-graph (*Figure 4G*). Only significant changes (p<0.05) of individual lipid *species* were included and compared to average *class* changes in *Figure 4G*. Monoacyl glycerides (MG) were excluded from this analysis as only one MG species was detected in only three of the seven syt7 and syt9 granule samples (see Supplemental Data Tables for lipidomic data and analysis).

## Protein purification

Syntaxin-1a (1-288 full length construct), SNAP-25, Munc18, Munc13, and complexin-1 from *Rattus norvegicus* were expressed in *Escherichia coli* strain BL21(DE3) cells under the control of the T7 promoter in the pET28a expression vector and purified as described previously (*Kreutzberger et al., 2016*; *Kreutzberger et al., 2017a*; *Kreutzberger et al., 2019*). SNAP-25 was quadruply dodecylated through disulfide bonding of dodecyl methanethiosulfonate (Toronto Research Company, Toronto, Ontario) to its four native cysteines (*Kreutzberger et al., 2016*).

To produce the iGluSnFr.A184S fluorescence glutamate sensor, plasmid containing the GltI-cpsfGFP variant (pRSET) of the SF-iGluSnFR.A184S (kindly provided by Jonathan S Marvin) (*Marvin et al., 2013*; *Marvin et al., 2018*) was transformed into *E. coli* BL21(DE3) cells (NEB). Protein expression was induced in TB-medium supplemented with 0.3 mM IPTG and 40 μg/ml ampicillin and incubated overnight at 25˚C. Cells were resuspended in ice cold extraction buffer (20 mM HEPES, 500 mM NaCl and 8 mM imidazole, pH 7.4) and lysed by sonication after incubation with lysozyme, DNAse, MgCl$_2$ and EDTA-free Protease inhibitor (Roche) for 1 h, 4 ˚C. Cell debris was removed by centrifugation and the protein was subsequently purified using Ni$^{2+}$-nitrilotriacetic acid beads (Qiagen GmbH) in 20 mM HEPES buffer containing 1 M NaCl and 150 mM imidazole, pH 7.4 followed by ion exchange chromatography (HiTrap QFF, GE Healthcare).

## Formation of planar supported bilayers with reconstituted plasma membrane SNAREs

Planar supported bilayers with reconstituted plasma membrane SNAREs were prepared by Langmuir-Blodgett/vesicle fusion technique as described in previous studies (*Domanska et al., 2009*; *Kalb et al., 1992*; *Wagner and Tamm, 2001*). Quartz slides were cleaned by dipping in 3:1 sulfuric acid:hydrogen peroxide for 15 min using a Teflon holder. Slides were then rinsed in milli-Q water. The first leaflet of the bilayer was prepared by Langmuir-Blodgett transfer onto the quartz slide using a Nima 611 Langmuir-Blodgett trough (Nima, Conventry, UK) by applying the lipid mixture of 70:30:3 bPC:Chol:DPS from a chloroform solution. Solvent was allowed to evaporate for 10 min, the

monolayer was compressed at a rate of 10 cm$^2$/min to reach a surface pressure of 32 mN/m. After equilibration for 5 min, a clean quartz slide was rapidly (68 mm/min) dipped into the trough and slowly (5 mm/min) withdrawn, while a computer maintained a constant surface pressure and monitored the transfer of lipids with head groups down onto the hydrophilic substrate.

Plasma membrane SNARE containing proteoliposomes with a lipid composition of bPC:bPE:bPS:Chol:PI:PI(4,5)P$_2$ (25:25:15:30:4:1) were prepared by mixing the lipids and evaporating the organic solvents under a stream of N$_2$ gas followed by vacuum desiccation for at least 1 hr. The dried lipid films were dissolved in 25 mM sodium cholate in a buffer (20 mM HEPES, 150 mM KCl, pH 7.4) followed by the addition of an appropriate volume of synatxin-1a and SNAP-25 in their respective detergents to reach a final lipid/protein ratio of 3000 for each protein. After 1 hr of equilibration at room temperature, the mixture was diluted below the critical micellar concentration by the addition of buffer to the desired final volume. The sample was then dialyzed overnight against 1 L of buffer containing Biobeads, with one buffer change after ~4 hr. To complete the formation of the SNARE containing supported bilayers, proteoliposomes were incubated with the Langmuir-Blodgett monolayer with the proteoliposome lipids forming the outer leaflet of the planar supported membrane and most SNAREs oriented with their cytoplasmic domains away from the substrate and facing the bulk aqueous region (*Domanska et al., 2009*; *Kalb et al., 1992*; *Kiessling et al., 2017*; *Wagner and Tamm, 2001*). A concentration of ~77 µM total lipid in 1.2 mL total volume was used. Proteoliposomes were incubated for 1 hr and excess proteoliposomes were removed by perfusion with 5 mL of buffer (120 mM potassium glutamate, 20 mM potassium acetate 20 mM HEPES, 100 µM EDTA, pH 7.4).

## TIRF microscopy for reconstituted planar supported membrane fusion assay

Experiments examining single-granule docking and fusion events were performed on an Axiovert 35 fluorescence microscope (Carl Zeiss), with a 63x water immersion objective (Zeiss, numerical aperture, 0.95) and a prism-based TIRF illumination. The light source was an OBIS 532 LS laser or an OBIS 488 LX laser from Coherent Inc Fluorescence was observed through a 610-nm band-pass filter (D610/60, Chroma) by an electron multiplying charge coupled device (CCD) (DU-860E, Andor Technology). The electron multiplying CCD (EMCCD) was cooled to −70℃, and the gain was set at 200. The prism-quartz interface was lubricated with glycerol to allow easy translocation of the sample cell on the microscope stage. The beam was totally internally reflected at an angle of 72° from the surface normal, resulting in an evanescent wave that decays exponentially with a characteristic penetration depth of ~100 nm. An elliptical area of 250 µm x 65 µm was illuminated. The laser intensity, shutter, and camera were controlled by a homemade program written in LabVIEW (National Instruments).

Experiments triggering secretory vesicle fusion with calcium were performed on a Zeiss AxioObserver Z1 fluorescence microscope (Carl Zeiss), with objective and TIRF setups as described above. The light source was a 488 or 514 nm beamline of an argon ion laser (Innova 90C, Coherent), controlled through an acousto-optic modulator (Isomet), and a diode laser (Cube 640, Coherent) emitting light at 640 nm. The characteristic penetration depths were between 90 and 130 nm. An OptoSplit (Andor Technology) was used to separate the fluorescence from the two colors. Fluorescence signals were recorded by an EMCCD (iXon DV887ESC-BV, Andor Technology). The EMCCD camera was cooled to −70℃, and the electron gain was set at 200.

## Calcium-triggered single vesicle – planar supported membrane fusion assay

Planar supported bilayers containing syntaxin-1a:SNAP-25 (bulk phase-facing leaflet lipid composition of 25:25:15:30:4:1 bPC:bPE:bPS:Chol:PI:bPIP$_2$) were incubated with 0.5 µM Munc18 and 2 µM complexin-1. Secretory vesicles were then injected while keeping the concentrations of Munc18 and complexin-1 constant. Secretory vesicle docking was allowed to occur for ~20 min before the chamber was placed on the TIRF microscope and the microscope was focused on the planar supported membrane. Docking was quantified by counting bound vesicles and normalizing the numbers to those obtained under a reference condition (syx(183-288):dSN25) on the same day. Fluorescence

from the sample was recorded while buffer containing 100 µM calcium was injected with a soluble Alexa647 dye in the buffer to monitor the arrival of calcium at the observation site.

## Fluorescence correlation spectroscopy

Steady state FCS measurements were acquired on a Zeiss LSM 880 confocal microscope (Oberkochen, Germany) equipped with a 40x/1.2 M27 W Korr C-Apochromat objective. Sample temperature was monitored using a Thor Labs (Newton, New Jersey) TSP01 external temperature probe. Confocal volume was determined using Rhodamine B with a known diffusion coefficient of $4.50 \times 10^{-5}$ cm$^2$s$^{-1}$ at 25.0°C (*Gendron et al., 2008*) diluted in ultra-pure water. Samples containing GFP were excited at 514 nm at an average power of 10.5 µW measured at the objective with an external Thor Labs PM16-120 using an argon laser and fluorescence emission was detected from 526 to 695 nm using a GaAsp PMT spectral detector. Twenty steady state FCS correlation curves were collected for 30 s intervals for each sample, that were then averaged and fit to a one-state three-dimensional diffusion model incorporating blinking,

$$G(\tau) = 1 + \left( \sum_{i=1}^{3} \frac{\frac{f_i * n_i^2}{\sum_{i=1}^{3} (f_i * n_i)^2}}{\left(1 + \frac{\tau}{\tau_{d,i}}\right) * \left(1 + \frac{\tau}{\tau_{d,i}} * \frac{\omega_r^2}{\omega_z^2}\right)^{\frac{1}{2}}} \right) * \left( 1 + \frac{T_b * e^{\frac{\tau}{\tau_b}}}{1 - T_b} \right) \quad (1)$$

using Zeiss's Zen Black 2.3 FCS software extension pack where $f$ is the fraction of molecules, $n$ is the molecular brightness, $\tau_d$ is the diffusion time, $\omega_r$ is the lateral focus radius, $\omega_z$ is the axial focus radius, $T_b$ is the blinking fraction and $\tau_b$ is the blinking relaxation.

FCS data for cytokine-treated granules contained an additional fast component consistent with the presence of a triplet state (*Figure 1—figure supplement 3*). To fit these datasets, a linearly independent triplet state correction was added to *Equation 1* to yield a one-state three-dimensional diffusion model incorporating both blinking and triplet state dynamics,

$$G(\tau) = 1 + \left( \sum_{i=1}^{3} \frac{\frac{f_i * n_i^2}{\sum_{i=1}^{3} (f_i * n_i)^2}}{\left(1 + \frac{\tau}{\tau_{d,i}}\right) * \left(1 + \frac{\tau}{\tau_{d,i}} * \frac{\omega_r^2}{\omega_z^2}\right)^{\frac{1}{2}}} \right) * \left( 1 + \frac{T_b * e^{\frac{\tau}{\tau_b}}}{1 - T_b} \right) * \left( 1 + \frac{T_t * e^{\frac{\tau}{\tau_t}}}{1 - T_t} \right) \quad (2)$$

where $T_t$ is the triplet state fraction and $\tau_t$ is the triplet state relaxation time. More complex fitting models incorporating rotational diffusion were considered and applied with no appreciable change in diffusion coefficient.

Temperature was monitored throughout the experiment and correlated to each run to adjust for thermal fluctuations. The Vogel equation was used to account for changes in viscosity ($\eta$) as a function of temperature in Kelvins,

$$\eta(T) = e^{\left(A + \frac{B}{C+T}\right)} \quad (3)$$

where A, B and C are fit parameters -3.72, 578.919 and -137.546, respectively. The radius of hydration ($R_h$) was calculated using the Stokes-Einstein relation for each 20-run sample collection,

$$R_h = \frac{kT}{6\pi\eta(T)D(T)} \quad (4)$$

where the viscosity ($\eta$) and diffusion coefficients (D) are both functions of temperature in Kelvin. A minimum of three independent 20x30 second runs were performed for each sample. This triplicate was collected on minimally 3 independent days where confocal volume and temperature changes were accounted for. Final averaging of data was performed across radius of hydration $R_h$ measurements to account for temperature and viscosity differences across samples. The reported $R_h$ values correspond to a minimum of 20x30 second runs averaged three times for three independently run samples resulting in an average hydrodynamic radius comparison over 180 trial runs.

## Cryo-electron microscopy

For each sample, 3.5 µl of purified secretory vesicles were applied to C-flat holey carbon grids (Electron Microscopy Sciences) then manually blotted and plunge-frozen into liquid ethane. Images were recorded at a magnification of 25,000X under low electron-dose conditions using a Tecnai F20 electron microscope operating at 120kV with a 4096 × 4096 pixel CCD camera (Gatan, Pleasanton, CA). Diameters of vesicles in micrographs were measured manually in Fiji (*Schindelin et al., 2012*).

## Acknowledgements

This work was supported by NIH grants P01 GM072694 to LKT and R01 R01 DK091296 to JDC. Members of the Tamm lab are acknowledged for numerous helpful discussions and we thank Dr. Tomas Kirchausen for helpful comments on the manuscript.

## Additional information

### Funding

| Funder | Grant reference number | Author |
| --- | --- | --- |
| National Institutes of Health | P01 GM072694 | Lukas K Tamm |
| National Institutes of Health | R01 DK091296 | J David Castle |

The funders had no role in study design, data collection and interpretation, or the decision to submit the work for publication.

### Author contributions

Alex J B Kreutzberger, Conceptualization, Formal analysis, Validation, Investigation, Visualization, Methodology, Writing - original draft; Volker Kiessling, Conceptualization, Resources, Data curation, Software, Formal analysis, Validation, Investigation, Methodology, Writing - review and editing; Catherine A Doyle, Noah Schenk, Clint M Upchurch, Margaret Elmer-Dixon, Amanda E Ward, Julia Preobraschenski, Syed S Hussein, Weronika Tomaka, Patrick Seelheim, Iman Kattan, Megan Harris, Binyong Liang, Investigation; Anne K Kenworthy, Norbert Leitinger, Supervision, Validation; Bimal N Desai, Arun Anantharam, Supervision; J David Castle, Conceptualization, Formal analysis, Supervision, Funding acquisition, Validation, Methodology, Writing - review and editing; Lukas K Tamm, Conceptualization, Supervision, Funding acquisition, Writing - review and editing

### Author ORCIDs

Alex J B Kreutzberger https://orcid.org/0000-0002-9774-115X
Volker Kiessling http://orcid.org/0000-0002-9388-5703
Bimal N Desai http://orcid.org/0000-0002-3928-5854
J David Castle https://orcid.org/0000-0002-5177-4749
Lukas K Tamm https://orcid.org/0000-0002-1674-4464

### Decision letter and Author response

Decision letter https://doi.org/10.7554/eLife.62506.sa1
Author response https://doi.org/10.7554/eLife.62506.sa2

## Additional files

### Supplementary files

• Transparent reporting form

### Data availability

All data generated or analysed during this study are included in the manuscript and supporting files.

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
