## [Decision Letter]

**Acceptance summary:**

This paper shows that insulinoma cells have two distinct populations of insulin secretory granules that differ in their fusion properties, their size, their protein and lipid composition, and their susceptibility to treatments that mimic type I or type II diabetes (cytokines or palmitate). These findings provide a major advance in our understanding of the fundamental biology of insulin granule formation and secretion, with important implications for understanding diabetes.

**Decision letter after peer review:**

Thank you for submitting your article "Distinct insulin granule subpopulations contribute to the secretory pathology of diabetes types 1 and 2" for consideration by *eLife*. Your article has been reviewed by three peer reviewers, and the evaluation has been overseen by Suzanne Pfeffer as the Senior and Reviewing Editor. The following individuals involved in review of your submission have agreed to reveal their identity: Frédéric Meunier (Reviewer #1); Michael Ailion (Reviewer #3).

The reviewers have discussed the reviews with one another and you will be pleased to learn that the story is provisionally accepted for publication and we welcome a revised version that addresses the queries listed below.

This is a very exciting paper which shows that insulinoma cells have two distinct populations of insulin secretory granules that differ in their fusion properties, their size, their protein and lipid composition, and their susceptibility to treatments that mimic type I or type II diabetes (cytokines or palmitate). These findings provide a major advance in our understanding of the fundamental biology of insulin granule formation and secretion, with important implications for understanding diabetes. Overall, the paper is very solid and the data are very convincing. Experiments are performed well and the manuscript is clearly written.

Please discuss in the text:

1) "It seems like it should have been key to look at the difference in proinsulin vs. insulin content in these subpopulations? Are we looking at immature vs. mature secretory granules? Cryo EM image from 1G doesn't look like the typical halo-ed MSG."

2) Purity of gradient question – iodixanol (optiprep) gradient does isolate LDCV, but our own experience (+published) show that other compartments do come up in the same fraction. So, are the synaptotagmins expressed in other compartments such as lysosomes etc, which then leads the question to how do you know that the fractions isolations between conditions contain the same number of insulin granules? Are we looking at a shift in the amount of lysosomes or some other contaminating compartment?

3) Use of palmitate treatment as T2D model without hyperglycaemia?

4) The selection of proteins blotted for Figure 5B is interesting, but more information about the selection of this panel would be good.

5) The title seems to promise big things but the singular use of the GRINCH/INS1 cell line is arguably not the best. Would be good to have finished on something simple, like data showing human islets have changes in their subpopulations with T2D, or even mouse islets + streptozatocin (T1D) show changes between these subpopulations.

---

## [Author Response]

[…] Please discuss in the text:1) "It seems like it should have been key to look at the difference in proinsulin vs. insulin content in these subpopulations? Are we looking at immature vs. mature secretory granules? Cryo EM image from 1G doesn't look like the typical halo-ed MSG."

We do not think that we are looking at mature vs. immature secretory granules in our experiments. First, the distinct lipid compositions of the two subpopulations make it unlikely that they have a precursor-product relationship. Second, the GFP western blots in Figure 5—figure supplement 1 (upper right panel) show that the ratios of proinsulin-GFP, intermediate-GFP and C-peptide-GFP are quite similar in control, IDsyt7, and IDsyt9 granules, which should not be the case if one were the precursor of the other. We are currently preparing a follow up manuscript that deals with the biogenesis of the two types of granules and there is no indication in that work either that we are dealing with mature vs. immature granules in the current work.

Haloed granules are typically seen by conventional EM using chemically fixed and heavy metal-stained specimens. The granules are almost certainly better preserved in our cryoEM images that do not use any chemical fixation or heavy metal staining. Even so, the granules also appear denser in the center than in the periphery in our cryoEM images as seen in the enlarged figure in Author response image 1.

**Author response image 1. sa2fig1:** 

2) Purity of gradient question – iodixanol (optiprep) gradient does isolate LDCV, but our own experience (+published) show that other compartments do come up in the same fraction. So, are the synaptotagmins expressed in other compartments such as lysosomes etc, which then leads the question to how do you know that the fractions isolations between conditions contain the same number of insulin granules? Are we looking at a shift in the amount of lysosomes or some other contaminating compartment?

The granules collected in fraction 9 from the iodixanol gradient may have some minor lysosomal contamination. We think it is unlikely that this contamination significantly impacts our results because we follow C-peptide-GFP, which is an authentic granule marker, in our in vitro fusion experiments both after diabetes-mimicking treatments and immunodepletion with synaptotagmin isoform antibodies. Further, the western blots for GFP shown in Figure 5—figure supplement 1 were obtained from samples that were loaded based on equal fraction of total of each sample. It is clear that there are very similar levels of GFP signal in each sample, which would be incompatible with extensive organelle contamination in only one of the granule fractions. We have added a statement in the subsection “Synaptotagmin isoforms define the calcium affinity of distinct insulin granule subpopulations” about a possible minor contamination of fraction 9 with other cellular components, but that this will unlikely influence our results and conclusions.

3) Use of palmitate treatment as T2D model without hyperglycaemia?

We agree that glucose plus palmitate is often considered the mimicking treatment for T2D. However, some publications just use palmitate, e.g. Hoppa et al., 2009. To be more accurate we now state in the Introduction that our treatment emphasizes the lipotoxicity component of T2D.

4) The selection of proteins blotted for Figure 5B is interesting, but more information about the selection of this panel would be good.

We are not sure what else the reviewers are looking for. In the subsection “Many proteins distribute selectively between granule subpopulations”, which describe these results, we start the description of each group of proteins that we analyzed with a short paragraph what these proteins do and why we included them in our analysis of the two types of granules.

5) The title seems to promise big things but the singular use of the GRINCH/INS1 cell line is arguably not the best. Would be good to have finished on something simple, like data showing human islets have changes in their subpopulations with T2D, or even mouse islets + streptozatocin (T1D) show changes between these subpopulations.

We agree that it would be very interesting to extend our current studies to human islets, which is one of our future goals. While GRINCH and INS832/13 reiterate a wide range of properties of β-cells in islets, there is potential for heterogeneity among islet cells as well as distinct islets, which may add non-trivial complexities in analysis that would still need to be overcome. Indeed, there is already precedence for differential modulation of calcium sensitivity related to exocytosis and cell maturity in β-cells (Huang et al., 2018). We have added a few sentences in the Discussion (sixth paragraph) to highlight such future directions. We have also changed the title of the manuscript to say that subpopulations “are implicated in” rather than “contribute to” the secretory pathology of T1D and T2D.